# Loss of polarity regulators initiates gasdermin-E-mediated pyroptosis in syncytiotrophoblasts

Khushali Patel[2,*], Jasmine Nguyen[1,*] , Sumaiyah Shaha[1] , Amy Brightwell[1], Wendy Duan[1] , Ashley Zubkowski[3] , Ivan K Domingo[2], Meghan Riddell[1,2]

**The syncytiotrophoblast is a human epithelial cell that is bathed in maternal blood on the maternal-facing surface of the human placenta. It therefore acts as a barrier and exchange interface between the mother and fetus. Syncytiotrophoblast dysfunction is a feature of pregnancy pathologies, like preeclampsia. Dysfunctional syncytiotrophoblasts display a loss of microvilli, which is a marker of aberrant apical–basal polarization, but little data exist about the regulation of syncytiotrophoblast polarity. Atypical PKC isoforms are conserved polarity regulators. Thus, we hypothesized that aPKC isoforms regulate syncytiotrophoblast polarity. Using human placental explant culture and primary trophoblasts, we found that loss of aPKC activity or expression induces syncytiotrophoblast gasdermin-E-dependent pyroptosis, a form of programmed necrosis. We also establish that TNF-α induces an isoform-specific decrease in aPKC expression and gasdermin-E-dependent pyroptosis. Therefore, aPKCs are homeostatic regulators of the syncytiotrophoblast function and a pathogenically relevant pro-inflammatory cytokine leads to the induction of programmed necrosis at the maternal–fetal interface. Hence, our results have important implications for the pathobiology of placental disorders like preeclampsia.**

## Introduction

The placenta is a fetally-derived transient organ that performs critical functions like gas and nutrient exchange, secreting pregnancy-supporting hormones, and acting as a physical barrier between the maternal and fetal compartments to establish and maintain a healthy pregnancy (1). The maternal-facing surface of the human placenta is covered by a single giant multinucleated epithelial cell called the syncytiotrophoblast (ST), which measures up to 10 m$^2$ by the end of gestation (2). The ST is a highly polarized epithelial cell with dense apical microvilli decorating its surface,

but it lacks lateral membranous barriers between nuclei. Thus, the ST represents a completely unique type of epithelial barrier. The ST cannot undergo mitosis, thus, it is maintained by the fusion of underlying proliferative-mononuclear progenitor cytotrophoblasts (CT). The ST is the primary cell-type responsible for carrying out the essential placental functions listed above; therefore, ST stress and dysfunction are key features of common pregnancy complications like preeclampsia and intrauterine growth restriction (3, 4, 5, 6).

Apical–basal polarity is the maintenance of the discrete structure and function of the apical and basal surface of a cell and is a hallmark feature of many epithelia, like the ST. In particular, the apical surface is dominated by microvilli, which are membrane protrusions supported by a core bundle of F-actin (7, 8). Critically, active maintenance of cell polarity is important to support microvillar structural integrity. For example, microvillar F-actin crosslinkage to the cell membrane by linker proteins from the ezrin, radixin, and moesin family must be maintained via phosphorylation (7, 9, 10). Specifically, phosphorylation at the highly conserved threonine-567 residue of ezrin (Thr-564 in radixin and Thr558 in moesin) is necessary for inducing a conformational change in these linker proteins that allows for their binding to both the actin cytoskeleton and the membrane (10). Microvilli functionally increase the surface area of a cell to facilitate gas and nutrient exchange, are a site of endocytosis and signal transduction, and may be the site of extracellular vesicle release in the ST (11, 12). Importantly, disruption of the ST microvillar structure has been reported in placentas from intrauterine growth restriction and preeclampsia pregnancies (3, 13, 14, 15, 16). However, no studies have directly examined the mechanisms regulating maintenance of ST polarity and microvilli.

Apical–basal polarity is governed by evolutionarily conserved protein complexes. One such functional module is the Par complex that includes the scaffolding protein partitioning defective-3 (Par-3), partitioning defective -6 (Par-6), and a Ser/Thr kinase, atypical PKC (aPKC). The Par complex localizes apically and maintains apical identity in multiple cell types (17, 18, 19). Ultimately localizing and

[1]Department of Physiology, University of Alberta, Edmonton, Canada   [2]Department of Obstetrics and Gynecology, University of Alberta, Edmonton, Canada   [3]Department of Biological Sciences, University of Alberta, Edmonton, Canada

Correspondence: mriddell@ualberta.ca
*Khushali Patel and Jasmine Nguyen contributed equally to this work

activating aPKC isoforms. We have shown that three isoforms of aPKC are expressed in ST: aPKC-ι, aPKC-ζ, and a novel N-terminal truncated *PRKCZ*-encoded isoform—aPKC-ζ III (20). APKC-ζ III lacks the N-terminal PB-1 domain required for Par-6 binding (20, 21). Hence, it is unclear if it can be fully activated because interaction with binding partners via the PB-1 domain is thought to be required for the full activation of aPKC isoform kinase activity (21, 22). Importantly, aPKC isoforms have both individual and redundant functions. As such, aPKC-ζ global knockout mice have no embryonic phenotype (23, 24), whereas aPKC-λ/ι global knockout mice are embryonic lethal at ~E9 with severe growth restriction and placental maternal–fetal interface malformation (25, 26). But, aPKC-ζ can partially compensate for aPKC- λ/ι knockout in mouse embryos (23), indicating that some aPKC- λ/ι functions can also be carried out by aPKC-ζ, but some are carried out by aPKC- λ/ι alone. Therefore, consideration for isoform-specific and total aPKC function is necessary. Bhattacharya et al found that placental specific and global knockout of aPKC-λ/ι led to a lack of murine placental labyrinthine zone development, the primary zone for maternal/fetal exchange in mice (25). They also found that knockdown of *PRKCI* in human trophoblast stem cells decreased their ability to form ST, thus revealing an important role for aPKC-ι in maternal–fetal interface formation (25). However, the function of all aPKC isoforms in the ST has not been addressed to date.

In this study, we sought to identify whether aPKC isoforms regulate apical polarization and microvillar organization of the ST. Our study reveals that aPKC isoform activity and expression are critical for the maintenance of the ST apical surface structure, integrity, and function in a regionalized manner. Moreover, we show that TNF-α regulates ST aPKC-ι expression, and that TNF-α exposure and loss of aPKC activity leads to regionalized loss of ST microvilli and apical membrane integrity because of the induction of non-canonical pyroptosis.

# Results

### aPKC isoforms and ezrin colocalize at the ST apical surface

Our initial studies examining aPKC isoforms in the human placenta revealed an apparent lack of anti-aPKC-ι or anti-aPKC-ζ/ζ III signal accumulation at the ST apical membrane in early first trimester placental samples (20). In other cell types, apical localization is required for aPKC's to regulate apical–basal polarity, as mentioned above. Thus, we first performed colocalization analyses using antibodies targeting aPKC-ι or aPKC-ζ isoforms (which we previously validated to recognize aPKC-ζ and aPKC- ζ III (20)) and an anti-ezrin antibody, a ST apical membrane marker in a third trimester placenta (27). Importantly, both aPKCs and ezrin have been shown to be expressed in the cytotrophoblast progenitor cells and the ST (20, 28). As stated in the introduction, the multinucleate and highly microvilliated ST rests above the mononucleate CT progenitors. Hence, the position of the ST can be identified in confocal image stacks as a multinucleate structure that lacks lateral F-actin and sits atop mononucleate F-actin-encircled CT and stromal cells (Fig 1A and B). The F-actin at the apical surface of the ST is highly convoluted because of the branching of the F-actin network within the microvilli, whereas the CT lack this highly complex F-actin structure in both the first trimester and term tissue, allowing for the discrimination between the ST and CT structures (Fig 1A and B).

When examining the localization of aPKCs within the ST, as observed previously (20), the anti-aPKC-ι signal was weak and largely diffuse within the ST from 4–6 wk of gestation (Fig S1A). There was also a lack of consistent apical accumulation of anti-ezrin signal at the ST apical surface despite a highly accumulated and complex pattern of anti-β-actin signal in the apical region (Fig S1A). By 7–8 wk of gestation, both anti-aPKC-ι and anti-ezrin signal had strong regional apical accumulation with significantly increased colocalization of the two signals compared with 4–6 wk of gestation, suggesting increased proximity of aPKC-ι and ezrin in the apical ST (Figs 1D and S1A). Similar apical localization and aPKC-ι:ezrin colocalization coefficients were quantified in 9–12-wk and 37–40-wk gestation ST (Fig 1C and D). Apical accumulation of anti-aPKC-ζ signal followed a similar trend, with a significant increase in the apical aPKC-ζ:ezrin colocalization coefficient by 7–8 wk of gestation (Figs 1F and S1B). The colocalization coefficient of these signals did not vary significantly at 9–12 wk or 37–40 wk of gestation (Fig 1E and F).

### Decreased aPKC activity disrupts ST ezrin abundance and activity

Having established that aPKC isoforms and ezrin strongly and consistently colocalize at the ST apical surface from 9 wk of gestation, we used floating placental explant culture and a myristoylated aPKC pseudosubstrate inhibitor (aPKC inhibitor), which blocks both aPKC-ι and aPKC-ζ activities (29), to test if aPKCs may regulate the ST apical membrane structure via ezrin Thr-567 phosphorylation, as observed during murine intestinal development (30). After 6 h of treatment, a significant decrease in the ST anti-phospho Thr-567 ezrin to total ezrin signal and the total anti-ezrin signal was observed in both 9–12 wk and 37–40 wk aPKC inhibitor versus control-treated explants (Figs 2A–E and S2). AKT was previously shown to regulate ezrin phosphorylation at Thr-567 in a trophoblastic cell line (31), so we also confirmed that aPKC inhibitor treatment did not alter the activity of AKT in 9–12-wk placental explants (Fig S3A and B). Thus, aPKC activity regulates ezrin Thr-567 phosphorylation and expression in ST. Meaning when aPKC activity is diminished, there is less ezrin available in the ST, and of the remaining ezrin, less of it is in the fully activated conformation because of a lack of Thr-567 phosphorylation.

### Loss of aPKC function and expression decreases ST apical F-actin

We also noted an appreciable decrease or change in the signal pattern of apical anti-β-actin signal with aPKC inhibitor treatment (Fig 2A). Therefore, we quantified the amount of ST apical F-actin in aPKC inhibitor-treated explants using phalloidin. APKC inhibitor led to a greater than 50% decrease of ST apical phalloidin signal intensity in both first trimester and term explants (Figs 2F–H and S4A). Areas without appreciable changes in apical–phalloidin signal were interspersed with regions with a clearly decreased signal. Significantly reduced apical phalloidin intensity was also observed after both 2 and 4 h of aPKC inhibitor treatment without the appearance of seemingly phalloidin-devoid regions (Fig S4B–D).

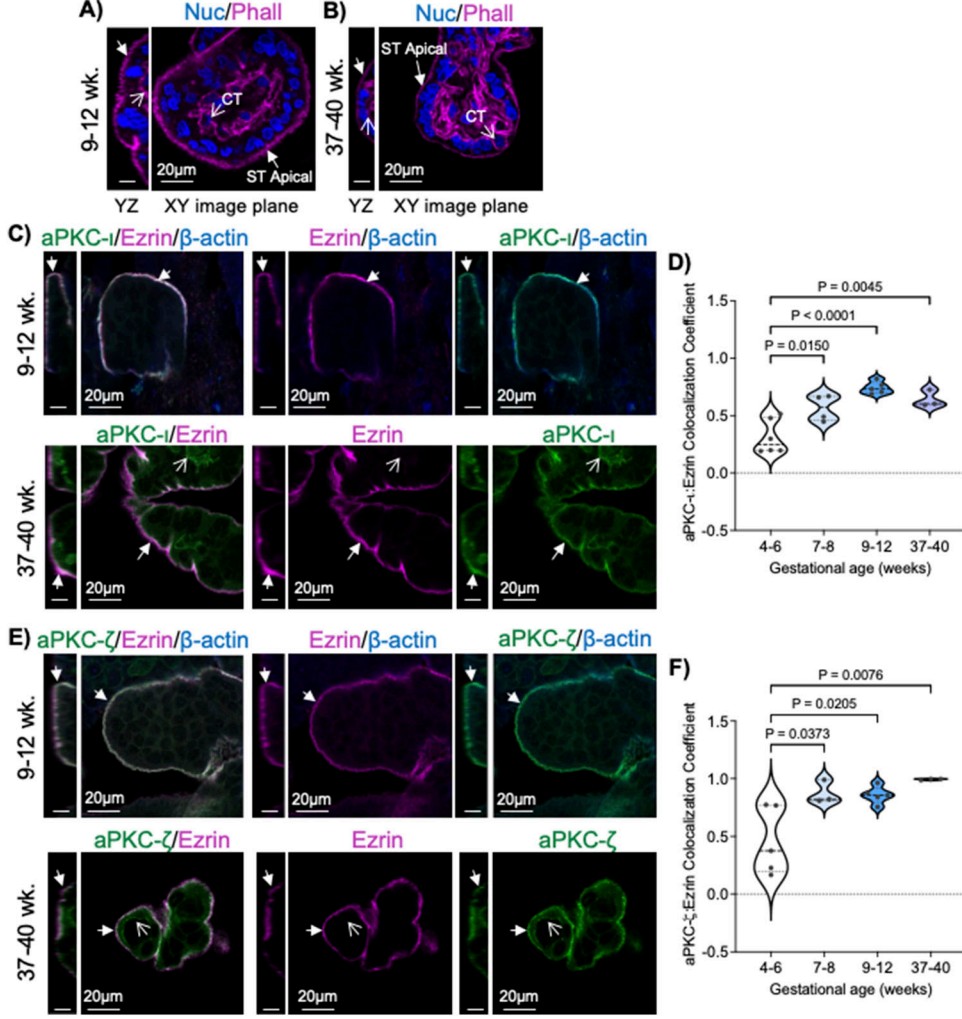

**Figure 1. aPKC-ι and aPKC-ζ strongly colocalize with ezrin at the ST apical surface in 9–12-wk and 37–40-wk placentas.** **(A)** Representative images for orientation of ST and CT position and structure in 9–12-wk placental tissue. **(B)** Representative images for orientation of ST and CT position and structure in 37–40 wk placental tissue. For both (A, B), filled in arrows indicate the ST apical (maternal blood facing) surface; open arrowheads indicate mononucleate CT; whole-mount tissue stained with phalloidin (magenta) and Hoescht 33342 (blue); left panels = YZ image plane, scale bar = 10 μm; right panel = XY image plane; All images from z-stacked confocal image acquisition. **(C)** Representative images of 9–12-wk (top panel) and 37–40-wk (bottom panel) placenta tissues stained with anti-aPKC-ι (green), anti-ezrin (magenta), and anti-β-actin (blue [top panels]) or anti-aPKC-ι (green) and anti-ezrin (magenta [bottom panels]); left panel = YZ plane, scale bar = 10 μm; right panel = XY plane; filled arrow heads = ST apical surface; open arrowheads = CT signal. **(D)** Summary data for quantitation of aPKC-ι and ezrin Pearson's colocalization coefficient at the ST apical surface at indicated gestational ages; n = 3–6; bold dashed line = median. **(E)** Representative images of 9–12-wk (top panel) and 37–40-wk (bottom panel) placental explants stained with anti-aPKC-ζ (green), anti-ezrin (magenta), and anti-β-actin (blue [top panels]) or anti-aPKC-ζ (green) and anti-ezrin (magenta [bottom panels]); left panel = YZ plane, scale bar = 10 μm; right panel = XY plane; Filled arrow heads = ST apical surface; Open arrowheads = CT. **(F)** Summary data for quantitation of ST apical aPKC-ζ and ezrin Pearson's colocalization coefficient at the indicated gestational ages; n = 3–5; bold dashed line = median; All analyses performed with one-way ANOVA with Tukey's post hoc test.

To address aPKC isoform-specific roles in ST, we performed siRNA-mediated knockdown. 9–12-wk gestation explants were treated with siRNA-targeting *PRKCI* and/or *PRKCZ*. Knockdown efficiency for aPKC-ι, aPKC-ζ, and aPKC-ζ III was determined by Western blot analyses of explant lysates (Fig 2I–P). Treatment with *PRKCI* siRNA, *PRKCZ* siRNA or both lead to significantly decreased ST apical phalloidin mean intensity (Fig 2Q and R). Importantly, as with aPKC inhibitor treatment, the appearance of seemingly phalloidin replete areas was regionally specific with siRNA knockdown. In addition, no change in the thickness of the ST could be observed using WGA lectin (Fig 2Q), a lectin that binds to the apical surface and cytoplasm of the ST (32, 33). A significant decrease in apical F-actin was observed with individual application of *PRKCI* siRNA and *PRKCZ* siRNA and no additive effect was observed with the addition of both (Fig 2R). Similar results were achieved with a second set of siRNAs targeting *PRKCI* or *PRKCZ* (Fig S5A–H). This suggests that reduced expression of a single isoform is sufficient to alter the ST apical actin cytoskeletal dynamics and that aPKC isoforms redundantly regulate ST apical membrane F-actin abundance.

## ST microvilli are lost with decreased aPKC activity

To confirm if loss of aPKC kinase activity leads to alterations in the ST apical membrane structure, we performed electron microscopy on control and aPKC inhibitor treated first trimester placental explants. Scanning electron micrographs (SEM) revealed a severe loss of microvilli at the ST apical surface with inhibitor treatment, apparent coalition of the few remaining microvilli, and a porous appearance of the apical membrane (Figs 3A and S6) in a region-specific manner. Transmission electron microscopy micrographs (TEM) confirmed the simplification of the apical surface structure apparent by SEM and revealed a ST-specific loss of cytoplasmic density, a high abundance of variably sized membrane coated vesicles, and swollen mitochondria (Fig 3B). Underlying cytotrophoblast progenitor cells did not have appreciable changes in any of these parameters and the basement membrane thickness between the cells did not vary between treatment groups.

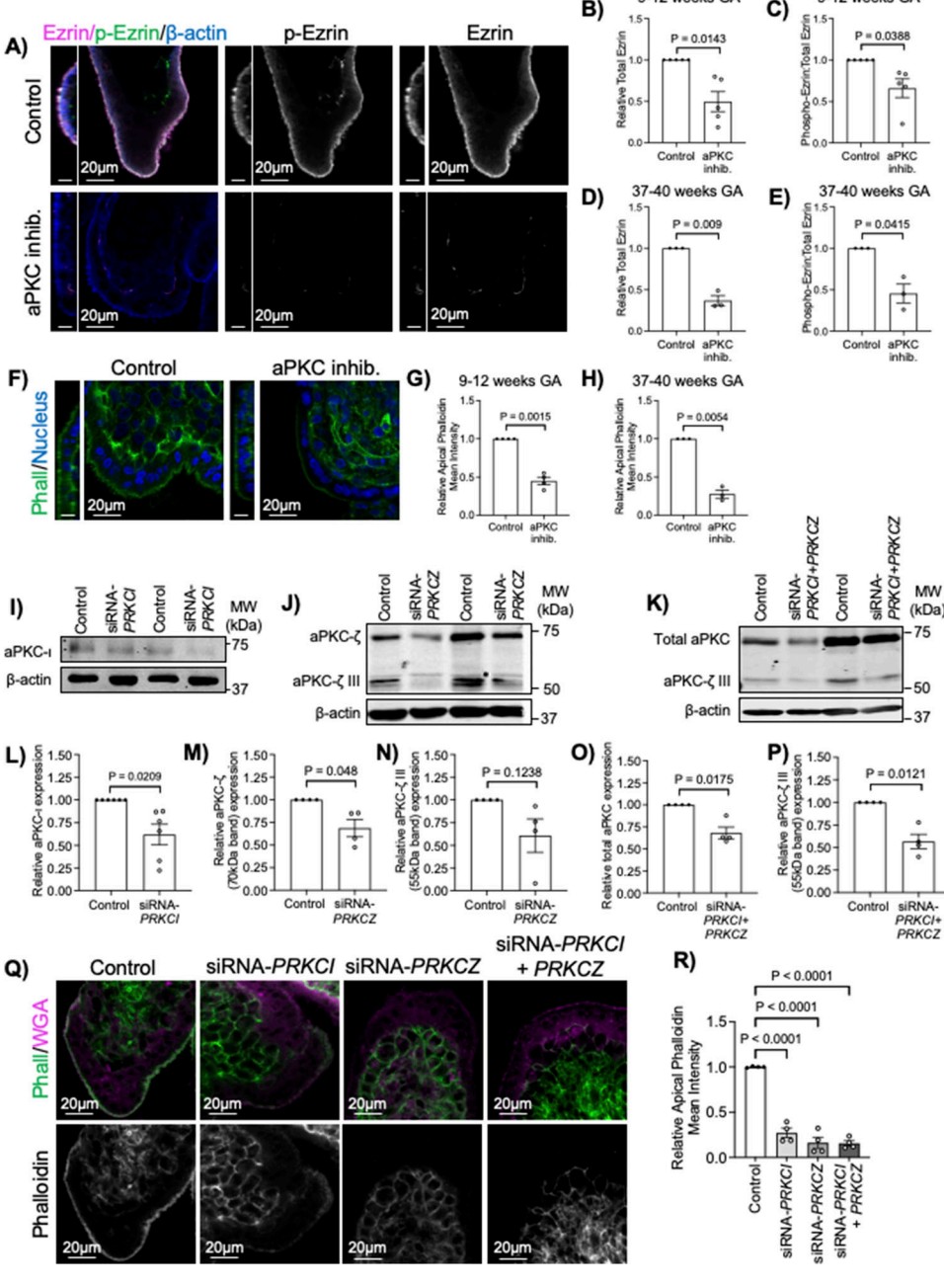

**Figure 2. Loss of aPKC kinase activity or expression alters apical ezrin activation and abundance and F-actin.**
**(A)** Representative images of 9–12-wk placental explants treated ± aPKC inhibitor for 6 h stained with anti-phospho(Thr567)-ezrin (green), anti-ezrin (magenta), and anti-β-actin (blue); left panel = YZ plane, scale bar = 10 $\mu$m; right panel = XY plane. **(B)** Summary data for quantitation of ST apical ezrin signal intensity in 9–12-wk explants; $n$ = 5. **(C)** ST apical phospho(Thr567)-ezrin relative to total ezrin signal intensity in 9–12-wk explants; $n$ = 5. **(D)** ST apical ezrin signal intensity in 37–40-wk explants; $n$ = 3. **(E)** ST apical phospho(Thr567)-ezrin relative to total ezrin signal intensity in 37–40-wk explants; $n$ = 3. **(F)** Representative images of phalloidin (green) and Hoechst-33342 (blue) in 9–12-wk placental explants treated ± aPKC inhibitor for 6 h; left panel = YZ plane, scale bar = 10 $\mu$m; right panel = XY plane. **(G)** Summary data for quantitation of ST apical phalloidin signal intensity in (G) 9–12-wk explants; $n$ = 4. **(H)** 37–40-wk explants; $n$ = 3; All above analyses performed using one-sample $t$ test; summary graphs mean ± SEM. **(I)** Representative Western blot with (I) anti-aPKC-ι and anti-β-actin after siRNA knockdown targeting *PRKCI*. **(J)** anti-aPKC-ζ and anti-β-actin after siRNA knockdown targeting *PRKCZ*. **(K)** anti-total-aPKC and anti-β-actin after siRNA knockdown targeting *PRKCI* and *PRKCZ*. **(L)** Summary data for quantitation of relative aPKC-ι expression normalized to scrambled control following siRNA knockdown targeting *PRKCI*; $n$ = 6. **(M)** aPKC-ζ expression normalized to scrambled control after siRNA knockdown targeting *PRKCZ*; $n$ = 4. **(N)** aPKC-ζ III expression normalized to scrambled control after siRNA knockdown targeting *PRKCZ*; $n$ = 4. **(O)** Total aPKC expression normalized to scrambled control after siRNA knockdown targeting *PRKCI* and *PRKCZ*; $n$ = 4. **(P)** aPKC-ζ III expression normalized to scrambled control after siRNA knockdown targeting *PRKCI* and *PRKCZ*; $n$ = 4. **(L, M, N, O, P)** statistics = one-sample $t$ test. **(Q)** Representative images of phalloidin (green) and WGA lectin (magenta) in 9–12-wk placental explants treated with isoform-specific siRNA for 24 h. **(R)** Summary data for quantitation of ST apical phalloidin signal intensity in 9–12-wk explants treated with aPKC isoform-specific siRNA; $n$ = 4. One-way ANOVA with Dunnett's post hoc test; summary graphs mean ± SEM.
Source data are available for this figure.

## Loss of aPKC expression and activity increases ST permeability

Because SEM and TEM imaging revealed a regionalized decrease in cytoplasmic density and an almost lace-like appearance in the apical membrane, we hypothesized that there is an increased permeability in ST with inhibition of aPKCs. Indeed, aPKC inhibitor treatment and *PRKCI + PRKCZ* siRNA knockdown both led to a ~3.5-fold increase in the uptake of 10,000 MW neutral dextran (Fig 3C–F)

compared with controls. *PRKCI* and *PRKCZ* siRNA-treated samples also displayed a significant 3.1 and 2.5-fold increase in sum dextran signal (Fig 3E and F). Importantly, this strong diffuse pattern of dextran uptake was restricted to areas of the ST lacking a visible continuous apical phalloidin signal (Fig 3C). Thus, decreased aPKC isoform kinase activity or expression disrupts ST apical membrane integrity and permeabilizes the ST to a neutrally charged 10,000 MW compound.

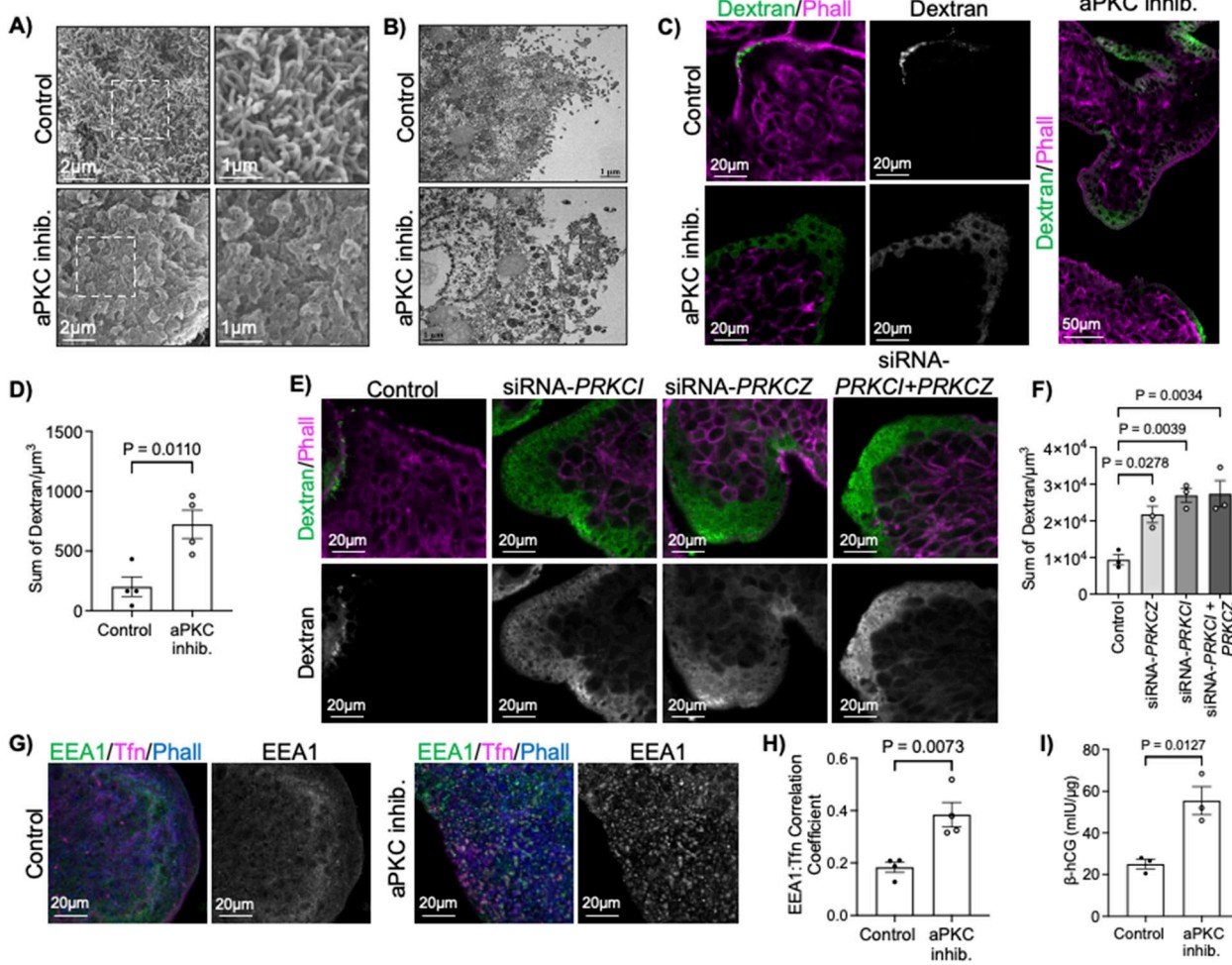

**Figure 3. aPKC inhibition leads to ST apical membrane simplification, decreased microvilli and cytoplasmic density, permeabilization of ST, and alteration of ST endocytic trafficking.**
**(A)** Representative SEM images of 9–12-wk placental explants treated with aPKC inhibitor for 6 h (representative of *n* = 3); right panels = isolated zoomed images of boxed area indicated on the left. **(B)** Representative transmission electron microscopy micrographs images of 9–12-wk placental explants treated with ± aPKC inhibitor for 6 h (representative of *n* = 3). **(C)** Representative images of 9–12-wk placental explants dextran-Texas Red (green) and phalloidin (magenta) after 2 h of treatment with aPKC inhibitor; left panels = merged image; middle panels = isolated dextran signal; right panel = merged dextran-Texas Red (green) and phalloidin (magenta) tile scan-stitched images of aPKC inhibitor-treated tissue. **(D)** Summary data for quantitation of the sum of dextran signals in 9–12-wk placental explants after 2 h of aPKC inhibitor treatment; *n* = 4; *t* test. **(E)** Representative images of 9–12-wk placental explants dextran-Texas Red (green) and phalloidin (magenta) after 24 h of treatment with scramble control or siRNA sequences targeting *PRKCI* and/or *PRKCZ*; top panels = merged image; bottom panels = isolated dextran signal. **(F)** Summary data for quantitation of the sum dextran signal in 9–12-wk placental explants after siRNA knockdown of *PRKCI* and/or *PRKCZ*; *n* = 3; one-way ANOVA with Dunnett's post hoc test. **(G)** Representative images of anti-EEA1 (green), transferrin-594 (magenta), and phalloidin (blue) in 9–12-wk placental explants after 2 h of aPKC inhibitor treatment; left panels = merged images; right panels = isolated EEA1 signal. **(H)** Summary data for quantitation of Global Pearsons correlation coefficient for EEA1:transferrin in 9–12-wk placental explants; *n* = 4; *t* test. **(I)** Summary data for quantitation of β-hCG concentration normalized to total protein levels in explant-conditioned medium ± aPKC inhibitor treatment; *n* = 3, *t* test. All summary graphs are mean ± SEM; all images in the single image plane of z-stack images.

## Disruption of aPKC function increases endosome size and uptake

In addition to the effects on the apical membrane, TEM also revealed the appearance of highly variably sized cytoplasmic membrane-coated vesicles (Fig 3B). In combination with the data showing altered F-actin abundance we hypothesized that loss of aPKC activity may lead to disrupted ST endocytic trafficking. Interestingly, when transferrin uptake assays were performed in first-trimester explants, we saw an increase in the colocalization of fluorescently-conjugated transferrin, a glycoprotein essential in the delivery of $Fe^{3+}$ into cells via transferrin receptor-mediated endocytosis (34), with anti-early endosome antigen-1 (EEA1), an endosomal marker protein (35), thus suggesting there is an increase in clathrin-mediated transferrin endocytosis with aPKC inhibitor treatment (Figs 3G and H and S7A–E). There were also enlarged areas of the anti-EEA1 signal at the ST apical surface in first-trimester explants treated with aPKC inhibitor (Figs 3G and S7B and D). Interestingly, these enlarged EEA1 signals were not exclusively associated with regions of the ST with a visibly decreased phalloidin signal. These data suggest that inhibition of aPKCs dysregulates ST clathrin-mediated endocytosis and leads to an increased endocytic activity and/or enlargement of ST endosomes in the apical compartment.

We reasoned that the altered vesicular trafficking and increased cellular permeability induced by aPKC-inhibitor treatment may also lead to increased release of ST-derived factors. The ST produces several pregnancy-sustaining hormones, including the hormone human chorionic gonadotropin (hCG). Therefore, we quantified the release of the β-hCG subunit via ELISA in an explant-conditioned medium from control and aPKC inhibitor-treated samples. As expected, aPKC inhibitor led to a nearly threefold increase of β-hCG in the explant-conditioned medium (Fig 3I). In summary, aPKC isoforms regulate the integrity of the apical membrane, vesicular trafficking within the ST, and structure of the apical compartment, likely via the regulation of F-actin and ezrin abundance.

### TNF-α decreases aPKC expression, apical F-actin, and induces ST permeability

TNF-α has been shown to decrease the expression of aPKC-ι (36), and is known to be elevated in the maternal circulation in established placental pathologies (37, 38) where regionalized loss of microvilli, reduced cytoplasmic density, and hyper-vesicular cytoplasm have also been observed in the ST (3, 14, 16). Hence, we hypothesized that TNF-α would also decrease the expression of aPKC isoforms in the ST. Interestingly, we found that treatment of in vitro-differentiated primary ST with TNF-α led to a dose-dependent significant decrease in aPKC-ι expression only (Fig 4A–D). In addition, both 0.1 and 10 ng/mL doses of TNF-α resulted in a regionalized but overall significant decrease in apical phalloidin in both first trimester and term explants (Figs 4E–G and S8B and C), like aPKC-inhibitor treatment or aPKC isoform siRNA knockdown. As expected, SEM revealed a regionalized severe loss of microvilli at the ST apical surface with TNF-α treatment and a similar porous or lace-like appearance of the apical membrane (Figs 4H and S8A) like with aPKC-inhibitor treatment. To confirm if this led to ST permeabilization, we performed dextran uptake assays that revealed a diffuse and significantly increased dextran signal in ST compared with control in both first trimester and term TNF-α-treated explants (Figs 4I–K and S8B). In addition, when explants were imaged by brightfield microscopy for 24 h, discrete areas of membrane blebbing on the placental surface were seen starting 12 h after treatment with TNF-α (Fig 4L and Video 1 and Video 2). Thus, TNF-α induces an isoform-specific dose-dependent decrease in ST aPKC-ι expression and simplification, permeabilization, and blebbing of the ST apical surface.

### TNF-α and loss of aPKC activity induce ST pyroptosis

Loss of aPKC expression/activity or exposure to TNF-α induces multiple forms of cell death, including apoptosis and pyroptosis, in other cell types (39, 40, 41, 42, 43, 44, 45). Pyroptosis is a pro-inflammatory form of programmed necrosis characterized by the appearance of membrane pores because of the cleavage and oligomerization of gasdermin family proteins (43, 46). Gasdermin pore formation leads to the permeabilization of the membrane to low-molecular weight compounds (47), and the release of mature IL-1β and other damage-associated molecular patterns (46, 48, 49). In contrast to apoptosis, which leads to the orderly disassembly of the nucleus and characteristic apoptotic nuclear disassembly (50),

pyroptotic nuclei do not undergo orderly nuclear disassembly, even after cell lysis (51). Because TNF-α and aPKC inhibitor treatment both induced regionalized formation of pore-like structures and increased permeability in ST but nuclear disassembly was only ever observed in the stromal cell compartment (Fig S9A), we hypothesized that the treatments were inducing ST pyroptosis. Interestingly, when IL-1β levels were tested in the explant-conditioned medium, a significant increase was only observed in first trimester explant-conditioned medium, but not in term explant-conditioned medium for both aPKC inhibitor and TNF-α (Fig 5A–D). The ST has previously been shown to express gasdermin-D (52); therefore, we stained TNF-α-treated explants with an anti-cleaved (Asp275) gasdermin-D (cl-GSDMD) antibody expecting to see an increase in cl-GSDMD signal at the ST apical membrane. In contrast to our hypothesis, no clear anti-cl-GSDMD signal could be observed within the ST, despite signal in both control and TNF-α treated cells in the villous core (Fig 5E). A similar staining pattern was observed in aPKC inhibitor-treated explants (Fig S9B). No cl-GSDMD signal was detected in the tissue that was fixed without culture, whereas a stromal cl-GSDMD signal was observed in an explant-cultured tissue from the same donor (Fig S10A–C). Gasdermin-E is a second gasdermin family member known to be expressed in the placenta (46). We first confirmed that it is expressed in the ST at both 9–12 wk and 37–40 wk of gestation using an anti-gasdermin-E antibody (Figs 5F and G and S11A–C), where a punctate signal pattern was observed in discrete areas of the ST, and sporadic cytotrophoblast progenitor cells at all gestational ages examined (Fig S11B). Interestingly, TNF-α treatment led to an apparent aggregation of anti-GSDME signal into large high signal intensity punctae at the apical border of the accumulated dextran signal in first trimester explants (Fig 5F). A similar change in anti-gasdermin-E signal was also observed in aPKC inhibitor-treated first trimester explants (Fig S9C). In term explants, only rare large puncta of anti-gasdermin-E signal were observed in the apical region, though high-signal intensity puntae were observed in the basal region of the ST with both treatments (Figs 5G and S9D). To confirm if gasdermin-E cleavage occurs in the ST, in vitro differentiated ST were treated with TNF-α and prepared for Western blotting analysis. Both full-length gasdermin-E and p30 cleaved gasdermin-E bands were observed, with a twofold increase in p30 gasdermin-E in TNF-α-treated cells (Fig 5H and I). Caspase-3 has been shown to cleave gasdermin-E, allowing for oligomerization and pore formation (45, 53), therefore we used an anti-active caspase-3 antibody (cleaved caspase-3) to stain TNF-α and aPKC inhibitor-treated explants. No signal could be detected within the ST in any treatment group in both first trimester and term explants (Figs 5J and K and S9E and F), though a clear signal could be detected in stromal cells, especially in TNF-α-treated explants. These observations are therefore consistent with previous studies showing caspase-3 activity is restricted to the stroma and cyto-trophoblasts (54).

### ST pyroptosis is gasdermin-E dependent

To identify if gasdermin-E is necessary for the increased permeability to dextran observed with TNF-α and aPKC-inhibitor treatment, 9–12-wk gestation explants were treated with gasdermin-E (encoded by DNFA5) targeting siRNA and then stimulated with

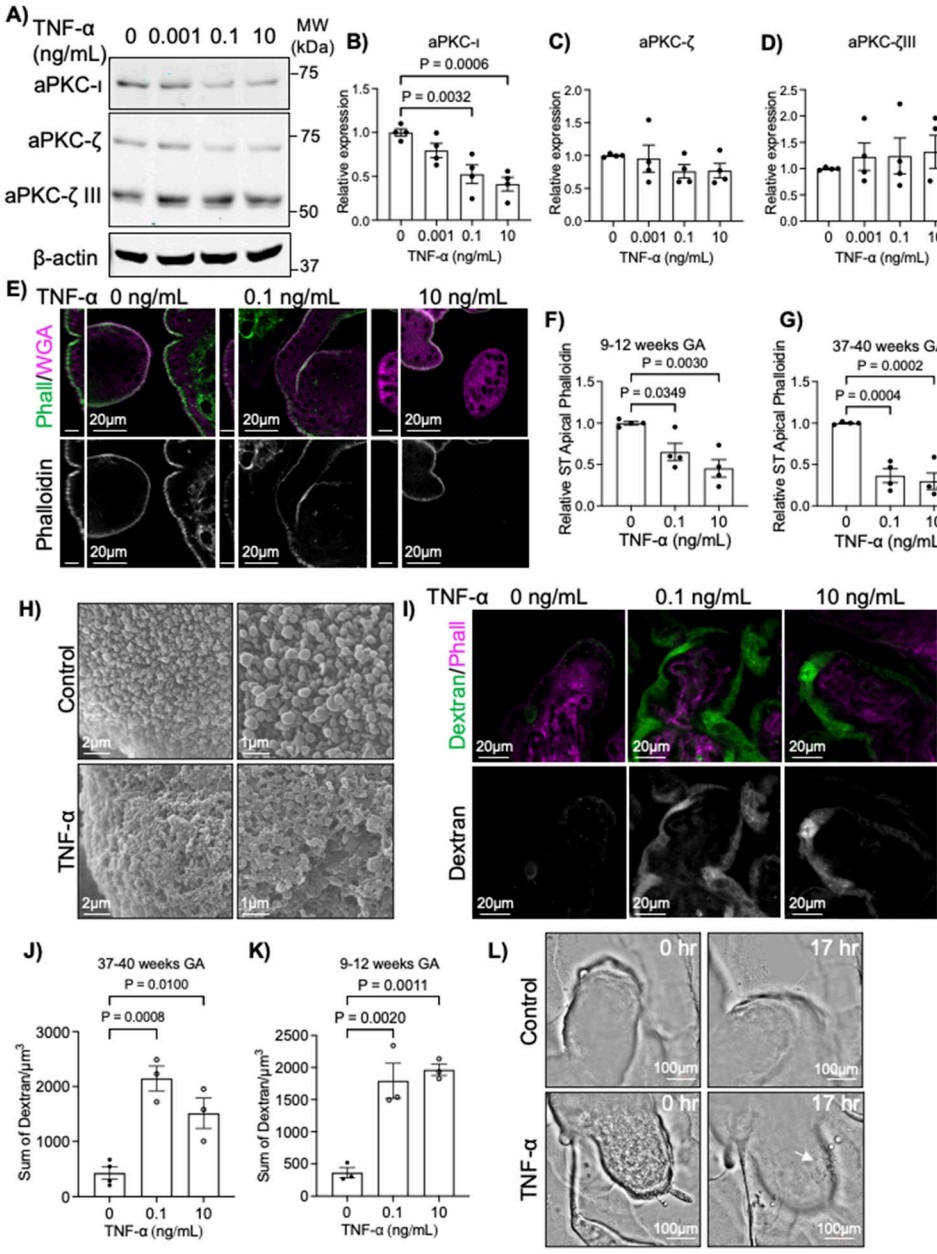

**Figure 4. TNF-α leads to a dose-dependent isoform-specific decrease in aPKC-ι expression, loss of apical F-actin, and permeabilization of ST.**
**(A)** Representative Western blot analyses of in vitro differentiated primary ST treated for 4 h with indicated doses of TNF-α, immunoblotted with anti-aPKC-ι, anti-aPKC-ζ, and anti-β-actin. **(B)** Western blot quantitation for aPKC-ι. **(C)** aPKC-ζ (70 kD form); and **(D)** aPKC-ζ III (55 kD form); $n$ = 4 patient-derived cells. **(E)** Representative images of phalloidin (green) and WGA lectin (magenta) in 9–12-wk placental explants treated with indicated doses of TNF-α for 6 h; top panels = merged image; bottom panels = isolated phalloidin signal; left panel = YZ plane, scale bar = 10 $\mu$m; right panel = XY plane. **(F)** Summary data for quantitation of ST apical phalloidin signal intensity in 9–12 wk tissue; $n$ = 4. **(G)** Summary data for quantitation of ST apical phalloidin signal intensity in 37–40 wk tissue; $n$ = 4. **(H)** Representative SEM images of 37–40 wk explants ±0.1 ng/mL TNF-α for 6 h; right panels = higher magnification images of the same samples; $n$ = 4. **(I)** Representative images of dextran-Texas Red (green) uptake and phalloidin (magenta) in 37–40-wk placental explants after 6 h of treatment with indicated doses of TNF-α; top panels = merged image; bottom panels = isolated dextran. **(J)** Summary data for quantitation of sum dextran in the ST in 37–40-wk tissue; $n$ = 3. **(K)** Summary data for quantitation of the sum dextran signal in the ST in 9–12-wk tissue; $n$ = 3. **(L)** Representative images of 9-wk placental explants at 0 and 17 h of treatment ±0.1 ng/mL TNF-α; arrow indicates a region of membrane blebbing. All analyses of one-way ANOVA with Dunnett's post-hoc test; summary graphs mean ± SEM; all images are single-image planes of z-stack images.
Source data are available for this figure.

either TNF-α or aPKC inhibitor. A significant decrease in gasdermin-E expression was observed after 48 h of treatment with siRNA by Western blotting analysis (Fig S12A and B). Critically, gasdermin-E knockdown blocked the increased dextran permeability induced by both TNF-α and aPKC inhibitor treatments (Fig 6A–D). Dimethyl fumarate (DMF) is an anti-pyroptotic agent that blocks the cleavage of both gasdermin-D and -E via succination (55). Therefore, we also sought to block TNF-α and aPKC inhibitor-induced pyroptosis by co-administration of the factors with DMF. As expected, DMF prevented TNF-α-induced ST permeability to dextran in both first trimester and term explants (Figs 6E and F and S13A and B). First trimester aPKC inhibitor-induced dextran accumulation could also be blocked by DMF (Fig 6G and H). Hence, the sum of these data reveal that TNF-α

and loss of aPKC kinase activity leads to the induction of ST pyroptosis, via a gasdermin-E-mediated pathway.

## Discussion

The ST inhabits an exceptional anatomical position as a fetal-derived cell bathed in maternal blood at its apical surface but attached to the fetal compartment at its basal edge. Here, we found that aPKC isoforms regulate the structure, permeability, and function of the ST apical surface in a spatially restricted manner by controlling the initiation of gasdermin-E-dependent pyroptosis. In addition, we established that the pro-inflammatory cytokine TNF-α

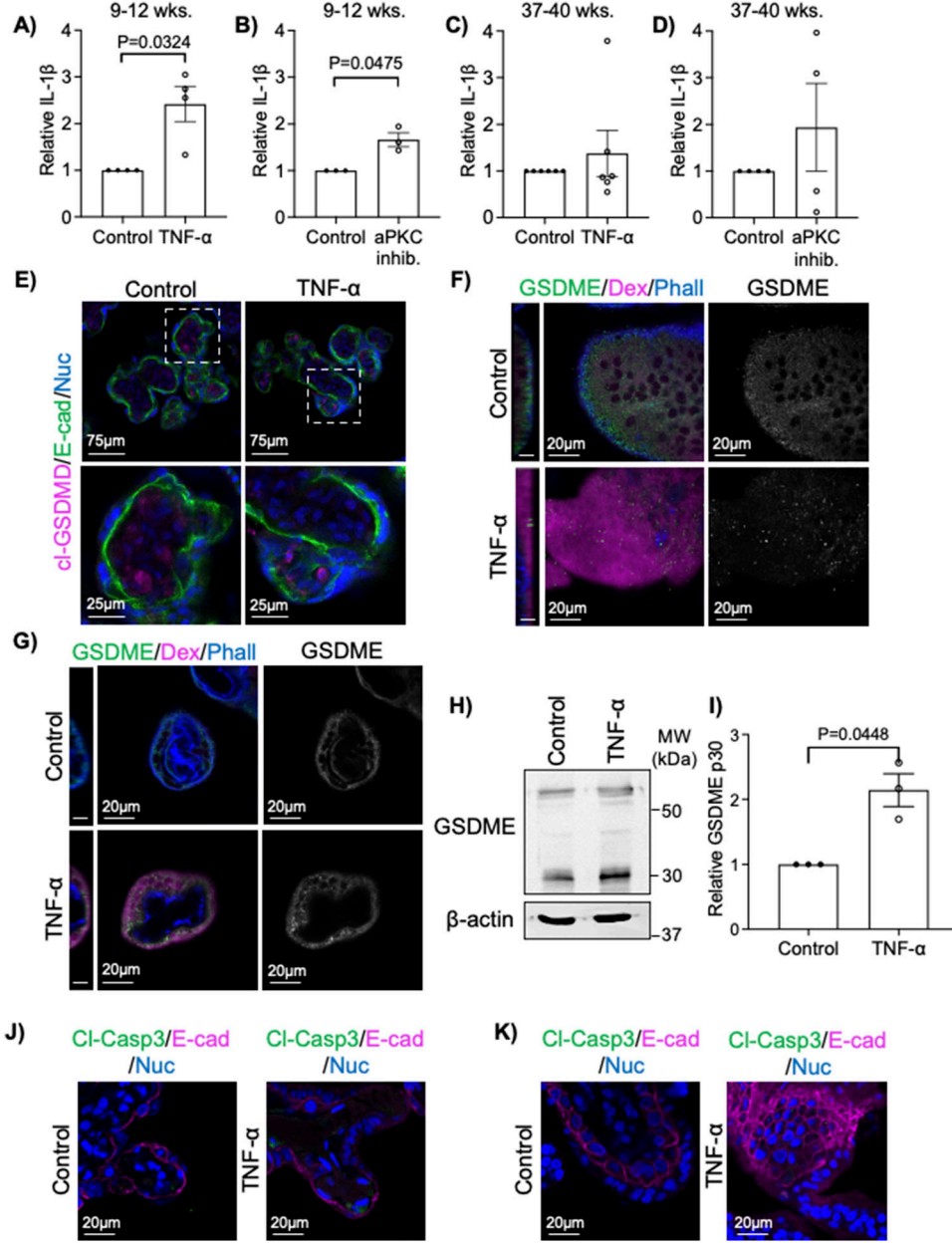

**Figure 5. TNF-α leads to cleavage of gasdermin-E.**
**(A)** IL-1β in 9–12-wk placental explant-conditioned medium after 6 h 0.1 ng/mL TNF-α stimulation; n = 4. **(B)** IL-1β in 9–12-wk placental explant-conditioned medium after 6 h aPKC inhibitor treatment; n = 3. **(C)** IL-1β in 37–40-wk placental explant-conditioned medium after 6 h 0.1 ng/mL TNF-α stimulation; n = 6. **(D)** IL-1β in 37–40-wk placental explant-conditioned medium after 6 h aPKC inhibitor treatment; n = 4. All analyses (A, B, C, D) one-factor t test. **(E)** Representative images of anti-cleaved gasdermin-D (cl-GSDMD, magenta); anti-E-cadherin (green), and Hoescht 33342-stained 37–40-wk placental explants ±0.1 ng/mL TNF-α for 6 h; bottom panels = higher magnification images of indicated areas. **(F)** Representative images of anti-gasdermin-E (GSDME, green), dextran-Texas Red (magenta), and phalloidin (blue) in 9–12-wk placental explants treated with ±0.1 ng/mL TNF-α for 6 h; left panel = YZ image plane of the merged image, scale bar = 10 μm; center panel = XY image plane of the merged image; right panel = XY gasdermin-E-isolated signal. **(G)** Representative images of anti-gasdermin-E (GSDME, green), dextran-Texas Red (magenta), and phalloidin (blue) in 37–40-wk placental explants treated with ±0.1 ng/mL TNF-α for 6 h; left panel = YZ image plane of the merged image, scale bar = 10 μm; center panel = XY image plane of the merged image; right panel = XY gasdermin-E-isolated signal. **(H)** Representative Western blot of in vitro differentiated primary ST treated with ±10 ng/mL TNF-α for 12 h. **(I)** Quantitation of Western blot analyses for gasdermin-E p30 normalized to total protein; n = 3 experimental replicates from n = 2 patient-derived cells; one-factor t test. **(J)** Representative images of anti-cleaved caspase 3 (cl-Casp3, green), anti-E-cadherin (magenta), and Hoescht 33342 (blue) signals in 37–40-wk placental explants ±0.1 ng/mL TNF-α for 6 h. **(K)** Representative images of anti-cleaved caspase 3 (cl-Casp3, green), anti-E-cadherin (magenta), and Hoescht 33342 (blue) signals in 9–12-wk placental explants ±0.1 ng/mL TNF-α for 6 h. All graphs mean ± SEM; all images are single-image planes of z-stack images.
Source data are available for this figure.

decreases ST aPKC-ι expression and profoundly alters ST apical structure and permeability leading to the initiation of pyroptosis. Therefore, our data suggest that aPKC isoforms are key regulators of ST homeostasis that can rapidly change in expression upon exposure to TNF-α leading to the release of a potent pro-inflammatory cytokine into maternal circulation.

Given the strong effects observed with the disruption of aPKC isoform expression and activity and that TNF-α leads to the isoform specific decrease in aPKC-ι expression, it will be important to identify the relevant regulators controlling aPKCs within the ST. It is presently unknown whether aPKC isoforms function as a part of the Par polarity complex in this cell type. Par-6 expression at the ST apical membrane from 12 wk of gestation has been shown (56),

which coincides with the gestational age range where we observed a significant signal accumulation for all aPKC isoforms at the apical surface (Fig 1). But presently, the expression and localization of Par-3 within the ST is unknown. Our data revealed profound regional disruption of the apical F-actin cytoskeleton, but also the appearance of very large EEA1-positive vesicular structures in regions where F-actin was not visibly diminished. Therefore, there are likely multiple pathways through which aPKCs facilitate ST function yet to be identified. Similarly, aPKC inhibitor treatment significantly decreased the ratio of activated (phospho-Thr567 ezrin) to total ezrin signal (Fig 2), indicating that aPKC kinase activity could be directly phosphorylating ezrin at the Thr-567 residue as previously demonstrated (30), and indirectly controlling ezrin abundance via an

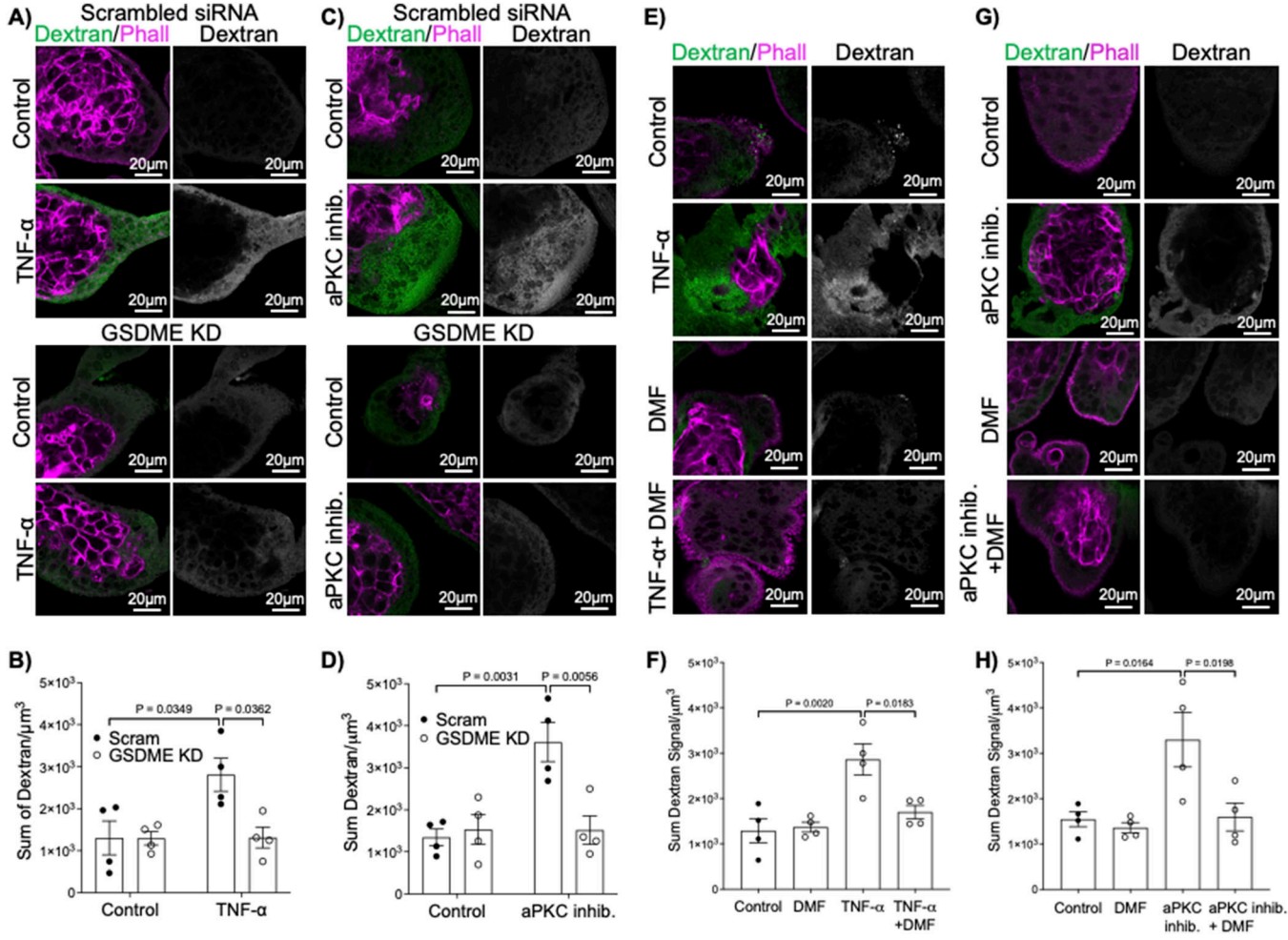

**Figure 6. GSDME knockdown blocks TNF-α and aPKC inhibitor induced ST permeability.**
**(A)** Representative images of dextran-Texas Red (green) and phalloidin (magenta) in 9–12-wk placental explants ± *DFNA5* siRNA and 6 h of treatment ±0.1 ng/mL TNF-α; left panels = merged images; right panels = isolated dextran signal. **(B)** Summary data for quantitation of sum dextran signal in the ST of 9–12-wk placental explants; *n* = 4. **(C)** Representative images of dextran-Texas Red (green) and phalloidin (magenta) in 9–12-wk placental explants ± *DFNA5* siRNA and 6 h of treatment ± aPKC inhibitor; left panels = merged images; right panels = isolated dextran signal. **(D)** Summary data for quantitation of sum dextran signal in the ST of 9–12-wk placental explants; *n* = 4. **(E)** Representative images of dextran-Texas Red (green) and phalloidin (magenta) in 9–12-wk placental explants after 6 h of treatment ± DMF, 0.1 ng/mL TNF-α, or both; left panels = merged images; right panels = isolated dextran signal. **(F)** Summary data for quantitation of sum dextran signal in the ST of 9–12-wk placental explants; *n* = 4. **(G)** Representative images of dextran-Texas Red (green) and phalloidin (magenta) in 9–12-wk placental explants after 6 h of treatment ± DMF, aPKC inhibitor or both; left panels = merged images; right panels = isolated dextran signal. **(H)** Summary data for quantitation of sum dextran signal in the ST of 9–12-wk placental explants in (G); *n* = 4; (B) two-way ANOVA with Sidak's multiple comparison test; (D) two-way ANOVA with Tukey's multiple comparison test; (F, H) one-way ANOVA with Sidak's multiple comparison test. All graphs mean ± SEM; All images are single XY image planes of z-stacked images.

additional pathway, to regulate ezrin homeostasis, microvillar structure, and maintenance. Intestinal cell regulation of aPKC-ι expression downstream of TNF-α is controlled by post-transcriptional mechanisms and ubiquitin-mediated degradation (36). Understanding how this is achieved in ST and whether other pro-inflammatory cytokines have similar effects will also be important in the future.

Our work also demonstrates that ST undergoes pyroptosis via a gasdermin-E-mediated pathway. These results are consistent with multiple studies that have identified ST-regionalized necrosis within the ST by electron microscopy (3, 16). These microvilli-replete ST regions have especially been identified in ST from preeclamptic pregnancies (16), where increased

placental mature IL-1β has also been reported (57). Placental mature-IL-1β was found to be predominantly produced by the ST (52) where the authors also observed a lack of cleaved-gasdermin-D signal, thereby supporting our conclusions that ST pyroptosis is not mediated by gasdermin-D. Though, it is presently unclear how gasdermin-E cleavage is occurring because no detectable caspase-3 activation was seen in ST in our experiments. The lack of caspase-3 activation is consistent with previous literature showing that cleaved-caspase-3 is restricted to the cytotrophoblast progenitors in vivo. (54) In addition, it is presently unknown how aPKC inhibition stimulates gasdermin-E-dependent pyroptosis in the ST. Therefore, further elucidation of the ST pyroptotic cascade is necessary.

Our data showing that the release of IL-1$\beta$ was not significantly altered by TNF-$\alpha$ treatment or aPKC inhibitor in term placental explants indicate that gestational age-dependent mechanisms may exist to control how this cytokine is released from the ST. This idea is consistent with the observations of Megli et al who found that constitutive NLRP3 inflammasome activation in the ST was significantly dampened in term compared with second trimester explants showing that there are gestational age-dependent regulatory changes controlling placental IL-1$\beta$ (52). IL-1$\beta$ can be released by gasdermin-dependent and -independent mechanisms (58, 59); therefore, it appears that gestational age-dependent alterations in the release pathways exist and require further investigation. In addition, elucidating other pyroptotic-initiating factors, and whether the mechanism of action has conserved regulatory points that could serve as therapeutic targets to block ST pyroptosis will also be an interesting future direction.

Our work also has clear implications for the pathobiology of placental disorders and infection during pregnancy. Presently, there are no data that we are aware of examining aPKC isoform expression in the placentas from pregnancy complications. Our results revealed that disruption of aPKC isoforms induces the rapid appearance of features characteristic of regions of the ST from preeclamptic pregnancies. Though preeclampsia is defined by the onset of maternal hypertension after the 20[th] wk of gestation and end organ failure (60, 61, 62), it is appreciated that the pathologic processes necessary for the development of the most severe form of the syndrome, early-onset preeclampsia, occur in early gestation (4). Interestingly, increased maternal first trimester circulating levels of IL-1$\beta$ have been reported in pregnancies that go on to develop early-onset preeclampsia (63). Therefore, our data showing that pyroptosis can be initiated in 9–12-wk placentas suggest that chronic initiation of this pathway could contribute to the progression of some forms of preeclampsia and should be directly tested in the future.

We also showed that regionalized membrane blebbing and the release of microvesicles occur during TNF-$\alpha$-stimulated ST pyroptosis in first trimester placental explants. Interestingly, proteins that form membrane pores, such as bacterial pore-forming toxins and gasdermin-D, elicit membrane-damage repair responses that function by increasing both endocytosis and exocytosis of membrane-coated vesicles to and from the damaged cell surface (64, 65). Therefore, the increased endocytic uptake we observed with aPKC inhibitor treatment may be a result of gasdermin-E-dependent membrane damage. In addition to increased endocytic uptake, we also saw the enlargement of EEA-1-positive vesicles after aPKC inhibitor treatment. Recently, the treatment of different cell lines with ionophores was shown to induce an enlargement in the size of endosomes without impeding their maturation (66). Gasdermin-D pores lead to $Ca^{2+}$ influx and $K^+$ efflux, therefore substantially disrupting the ionic balance of a cell, and this ionic imbalance results in a gasdermin-D pore-stimulated rise in exocytic release of microvesicles (67, 68). Hence, further work to establish if gasdermin-E pores result in similar ionic imbalance in the ST and the associated alterations in endosomal size, but not maturation, and exocytic release of different sizes of microvesicles and/or exosomes also deserve investigation. Importantly, an increase in ST-derived microvesicles and exosomes in maternal circulation of women with preeclampsia has been shown and is postulated to be a major mechanism leading to the development maternal vascular dysfunction in the syndrome (69, 70). Therefore, our work suggests that ST pyroptosis and the resultant induction of membrane blebbing, and possibly membrane repair mechanisms, could be a cause of increasing circulating levels of ST microvesicles, which could begin as early as the end of the first trimester.

# Materials and Methods

Please see Table 1 for a complete list of antibodies used in this study.

## Tissue collection

37–40-wk gestation and first trimester human placental samples were collected by methods approved by the University of Alberta Human Research Ethics Board (Pro00088052, Pro00089293). Term placental tissue was collected after cesarean delivery without labour from uncomplicated pregnancies. First trimester placental tissue was obtained from elective pregnancy terminations after informed consent from the patients.

## Floating placental explant culture and treatments

Placental samples were collected and rinsed in cold 1X PBS to remove blood. For term placentas, the tissue was cut from three central cotyledons, decidua and blood clots were removed, and the trimmed tissue was washed extensively in PBS to remove residual blood. For first trimester samples, placenta was identified, separated from decidua, blood clots were removed, and the tissue was washed extensively in PBS to remove residual blood. Both first trimester and term tissues were then cut into uniform 2-mm$^3$ explants, placed into 48-well plates and incubated overnight at 37°C 5% $CO_2$ in (IMDM; Gibco) supplemented with 10% (vol/vol) FCS (Multicell, Wisent Inc.; Saint-Jean Baptiste Canada) and penicillin–streptomycin (5,000 IU/mL; Multicell, Wisent Inc.). After overnight incubation, explants were washed in serum-free IMDM with 0.1% (wt/vol) BSA (Sigma-Aldrich) and incubated for 2–24 h at 37°C 5% $CO_2$ ± myristoylated aPKC pseudosubstrate inhibitor (5 $\mu$M; Product number 77749; Invitrogen) in IMDM + 0.1% BSA. For TNF-$\alpha$ treatments, after washing in a serum-free medium, the explants were preincubated for 30 min at 37°C 5% $CO_2$ in IMDM +0.1% BSA, then the medium was changed to IMDM +0.1% BSA ± TNF-$\alpha$ (0.1 or 10 ng/mL; Sigma-Aldrich) and incubated at 37°C 5% $CO_2$ for 6–24 h. Explants were then washed with cold PBS and fixed with 4% paraformaldehyde for 2 h on ice. For experiments performed with DMF (Sigma-Aldrich), explants were washed as above, and then preincubated for 30 min with solvent control (DMSO; 1:1,000) or DMF (25 $\mu$M). The medium was then changed and 0.1 ng/mL TNF-$\alpha$ or 5 $\mu$M myristoylated aPKC pseudosubstrate inhibitor was added ± DMF and incubated for 6 h at 37°C 5% $CO_2$. Explants were then washed with cold PBS and fixed with 4% paraformaldehyde for 2 h on ice. Technical triplicates were performed for all treatments.

**Table 1.  Antibodies used in study.**

| Target | Company | Product number | Clone | Lot | Validation |
|---|---|---|---|---|---|
| aPKC-ι | BD Biosciences | 610207 | 41/PKCλ | 7219769 | siRNA KD validated by Western blotting this study and reference ([71]) |
| aPKC-ι | Atlas Antibodies | HPA026574 | N/A | 24060 | Validation statement on the manufacturer's website |
| aPKC-ζ | Atlas Antibodies | HPA021851 | N/A | A1138749 | siRNA KD validated by Western blotting in this study and reference ([20]) |
| Total aPKC | Santa Cruz Biotechnologies | sc-17781 | H1 | C1122 | Knockdown validated reference ([72]) |
| AKT(pan) | Cell Signaling Technology | 2920 | 40D4 | 8 | None found |
| phospho Ser473 AKT | Cell Signaling Technology | 4060 | D9E | 25 | None found; in house validated with AKT inhibitor |
| Gasdermin-E | AbCam | ab215191 | EPR19859 | GR3279425 | Knockout validated by manufacturer |
| Gasdermin-E | Santa Cruz Biotechnologies | sc393162 | G-9 | K0121 | None found |
| cleaved GSDMD Asp275 | Cell Signaling Technology | 36425 | E7H9G | 3 | None found |
| β-actin | Cell Signaling Technology | 8457 | D6A8 | 7 | None found |
| β-actin | Cell Signaling Technology | 3700 | 8H10D | 18 | None found |
| Ezrin | Invitrogen | PA5-18541 | N/A | UH2820922 | None found |
| Phospho Thr567 Ezrin | Invitrogen | PA5-37763 | N/A | WA3173311 | None found |
| EEA1 | R&D Systems | MAB8047 | 871546 | CIIB030041 | None found |
| Cleaved Caspase-3 Asp175 | Cell Signaling Technology | 9661 | N/A | 47 | None found |
| E-cadherin | R&D Systems | MAB18381 | 180215 | JAT00221031 | None found |

## siRNA knockdown

siRNA knockdown was performed as previously reported ([20]). 9–12-wk placental explants were placed into a 48-well plate with IMDM supplemented with 10% (vol/vol) FCS, gentamicin (50 μg/mL; Gibco), and penicillin–streptomycin (5,000 IU/mL; Multicell, Wisent Inc.). siRNA sequences targeting *PRKCI* (final concentration 0.2 μM; ON-TARGETplus siRNA J-004656-07or J-04656-10; Dharmacon) and *PRCKZ* (final concentration 0.2 μM; ON-TARGETplus siRNA J-003526-17-0010 or J-003526-14; Dharmacon), siRNA targeting both *PRKCI* and *PRCKZ* or scrambled control (final concentration 0.2 μM; ON-TARGETplus Non-targeting Control Pool D-001810-10-20; Dharmacon) were added to the medium and incubated for 24 h siRNA sequences targeting *DFNA5* (final concentration 0.2 μM; hs.Ri.DFNA5.13.1; Integrated DNA Technologies) or negative control (final concentration 0.2 μM; Integrated DNA Technologies) were added to the medium and the explants were incubated for 48 h. After treatment, the explants were washed with cold PBS before fixation with 4% paraformaldehyde for 2 h on ice or collected in RIPA lysis buffer to perform Western blotting. For gasdermin-E (*DNFA5*) siRNA-treated explants, aPKC inhibitor and TNF-α treatment were added for 6 h after 48 h of culture with the siRNA.

## Dextran uptake assay

In the last 30 min of treatment with aPKC inhibitor, TNF-α or siRNA explants were incubated with 10,000 molecular weight (MW) neutral Dextran Texas Red (25 μg/mL; D1828; Invitrogen) for 25 min in IMDM + 0.1% BSA and washed with cold PBS before fixation with 4% paraformaldehyde for 2 h on ice.

## Transferrin endocytosis assay

After aPKC inhibitor treatment, explants were incubated with fluorescently conjugated human transferrin-594 (CF 594; 25 μg/mL; Biotium) for 40 min. The tissue was then washed extensively with cold 1X PBS and fixed in 4% paraformaldehyde for 2 h on ice.

## Primary trophoblast isolation, culture, and treatment

Term placental cytotrophoblasts were isolated according to previously published methods ([20], [73]). To obtain in vitro-differentiated syncytiotrophoblasts, isolated cytotrophoblast progenitor cells were seeded into six-well plates and cultured in IMDM + 10% FCS + penicillin–streptomycin and incubated for 4 h at 37°C 5% CO$_2$. Cells were then washed to remove non-adherent cells, the medium

changed to IMDM + 10% FCS + penicillin–streptomycin + 8-bromo-cAMP (10 μM; Sigma-Aldrich), and incubated overnight at 37°C 5% $CO_2$. The medium was changed to remove the 8-bromo-cAMP and the cells were incubated for a further 48 h 37°C 5% $CO_2$ (72 h in culture total).

TNF-α treatments were performed after differentiation into syncytiotrophoblasts (after 72 h in culture). The medium was removed, cells were washed, and the medium was replaced with IMDM + 0.1% BSA and cells were incubated for 30 min at 37°C 5% $CO_2$. The medium was then changed to IMDM + 0.1% BSA ± TNF-α at the indicated doses for 4–12 h. Cells were then washed, and protein lysates were prepared for Western blotting analysis.

### Western blotting

Samples were prepared by adding RIPA lysis buffer and protease inhibitor (1:100; P2714; Sigma-Aldrich) and protein concentration was determined using a BCA protein assay. Protein was loaded and run on SDS-polyacrylamide gels before being transferred onto nitrocellulose membranes. The membranes were probed with mouse anti-aPKC-ι (1:1,000; 610207; BD Biosciences), rabbit anti-aPKC-ζ (1:2,000; HPA021851; Atlas Antibodies), mouse anti-total aPKC (1:1,000; sc-17781; Santa Cruz Biotechnologies), mouse anti-AKT(pan) (1:2,000; #2920; Cell Signaling Technology), rabbit anti-phospho Ser473 AKT (1:2,000; #4060; Cell Signaling Technology), rabbit-anti-Gasdermin-E (1:10,000; ab215191; Abcam), and mouse or rabbit anti-β-actin (1:10,000 [both antibodies]; #8457; Cell Signaling Technology or #3700; Cell Signaling Technology) and fluorescent secondary antibodies. Secondary antibodies included Alexa Fluor donkey anti-mouse 680 (1:10,000; A28183; Invitrogen) and Alexa Fluor donkey anti-rabbit 800 (1:10,000; A21039; Invitrogen). Total protein quantification was performed by staining membranes with Fast-Green FCF (74). All blots were scanned on a Licor Odyssey scanner and quantitation was performed using the Licor Imaging Software with target protein band intensity normalized to total protein.

### Tissue staining and image analysis

After fixation-cultured explants or fixed un-cultured placental tissue was whole mount stained, the tissue was incubated with a blocking buffer (5% normal donkey serum and 0.3% Triton x100, 1% human IgG [Invitrogen] in PBS) followed by incubation with primary antibodies: anti-aPKC-ι (1:100; HPA026574; Atlas Antibodies); anti-aPKC-ζ (1:200, HPA021851; Atlas Antibodies); anti-ezrin (1:33; PA5-18541; Invitrogen); anti-phospho Thr567 ezrin (1:200; PA5-37763; Invitrogen); anti β-actin (1:250; #8457; Cell Signaling Technologies); anti-EEA1 (1:50; MAB8047; R&D Systems); anti-cleaved GSDMD Asp275 (1:500; #36425; Cell Signaling Technologies); anti-GSDME (1:50; sc393162; Santa Cruz Biotechnology); anti-cleaved caspase-3 Asp175 (1:400; #9661; Cell Signaling Technologies), anti-E-cadherin (1:400; MAB18381; R&D Systems) or biotinylated-WGA Lectin (1:1,000; Vector Biolabs) overnight, then washed and incubated with the appropriate secondary antibodies (Alexa Fluor, Invitrogen) and/or fluorescently conjugated phalloidin (1:400; iFluor 405 or iFluor594; AAT Bioquest). Hoechst 33342 (Invitrogen) was then added for 30 min. Tissue was then washed and mounted with imaging spacers.

All images were captured with a Zeiss LSM 700 confocal microscope using a Zeiss 20x/0.8 M27 lens or a Zeiss 63x/1.4 Oil DIC M27 lens. 10–20 μm Z-stack images with a 1 μm step-size were captured at 63x magnification. Three images per treatment were captured. Image capture was restricted to blunt-ended terminal projections with underlying cytotrophoblast progenitors in villi from first trimester and term samples.

### ELISA assays

Conditioned explant culture medium was collected from technical triplicates after 6 h incubation with treatments and centrifuged at 12,000$g$ for 10 min and the supernatant aliquoted then stored at −20°C. The tissue was washed with PBS then flash frozen and stored at −80°C until total protein was extracted and determined by BCA assay, as above. ELISAs were performed using a β-hCG ELISA kit (EIA-1911; DRG International) and IL-1β ELISA kit (DY401-05; R&D Systems). Plates were read using the Biotek Synergy HTX plate reader (Gen 5 software). The β-hCG and IL-1β values were interpolated using GraphPad PRISM 9 (Version 9.3.1) and normalized to the total protein from the explant the medium was produced by.

### Live cell imaging

9–12-wk gestation placental explants were washed and cut as detailed above. The tissue was then immersed in 1% low melting point agarose (Sigma-Aldrich) at 40°C for less than 30 s, placed in a 48-well plate, and overlaid with 5 μl of additional 1% low melting point agarose. Agarose was allowed to cool for 15 min at room temperature, then IMDM + 10% FCS + penicillin–streptomycin was added to each well and explants were incubated overnight at 5% $CO_2$ 37°C in a cell culture incubator. Treatments were then added as above, and plates were moved to a Zeiss Cell Discoverer 7 set to an atmosphere of 5% $CO_2$ 37°C. Oblique brightfield images were acquired with a 2-μm z-step every 30 min for 24 h with an Axiocam 712 mono camera using the Zen imaging software (version 3.4) and a Zeiss Plan-Apochromat 20x/0.7 autocorr lens. Movies and still images were processed in Zen imaging software.

### Statistical analysis

Statistical analyses were completed using GraphPad PRISM 9 (Version 9.3.1.) with α = 0.05 as the threshold for significance. Exact statistical methods used for individual experiments are contained in the figure legends. All graphs represent mean ± SEM. Statistical outliers were determined using a ROUT outlier analyses or Grubbs test in PRISM 9. Reported $n$ represents biological replicates (tissue/cells from different patient samples) and all experiments were repeated a minimum of three times.

## Supplementary Information

# Acknowledgements

We would like to thank Dr. Carolyn Jones for her feedback on the initial data, Maya Henriquez for the collection of term placental tissue, and all the families that donated tissue for the study. We would also like to thank Dr. Kacie Norton and Arlene Oatway of the University of Alberta Biological Sciences Imaging Facility for their expert technical advice for the electron microscopy sample preparation and imaging. This work was supported by funding from the Canadian Institutes of Health Research (MRC-167968) and the Women and Children's Health Research Institute and their generous donors: the Alberta Women's Health Foundation and the Stollery Children's Hospital Foundation. K Patel was supported by the MaTCH program stipend award. S Shaha is supported by an Alberta-Innovates Graduate Studentship. J Nguyen and W Duan were supported by summer studentships from the Women and Children's Health Research Institute, Alberta-Innovates, and NSERC.

## Author Contributions

K Patel: formal analysis and writing—original draft.
J Nguyen: formal analysis, methodology, and writing—original draft, review, and editing.
S Shaha: formal analysis, methodology, and writing—review and editing.
A Brightwell: formal analysis.
W Duan: formal analysis and writing—review and editing.
A Zubkowski: formal analysis and writing—review and editing.
IK Domingo: formal analysis and writing—review and editing.
M Riddell: conceptualization, resources, formal analysis, supervision, funding acquisition, methodology, and writing—original draft, review, and editing.

## Conflict of Interest Statement

The authors declare that they have no conflict of interest.

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
