## [Reviewer comments · Life Science Alliance]

Life Science Alliance

Loss of polarity regulators initiates gasdermin-E mediated pyroptosis in syncytiotrophoblasts

Khushali Patel, Jasmine Nguyen, Sumaiyah Shaha, Amy Brightwell, Wendy Duan, Ashley Zubkowski, Ivan Domingo, and Meghan Riddell

DOI: <https://doi.org/10.26508/lsa.202301946>

Corresponding author(s): Meghan Riddell, University of Alberta

Review Timeline:	Submission Date:	2023-01-24
	Editorial Decision:	2023-02-20
	Revision Received:	2023-06-19
	Editorial Decision:	2023-07-06
	Revision Received:	2023-07-10
	Accepted:	2023-07-12

Scientific Editor: Novella Guidi

Transaction Report:

February 20, 2023

Re: Life Science Alliance manuscript #LSA-2023-01946-T

Dr. Meghan Riddell
University of Alberta
Obstetrics and Gynecology
227 Heritage Medical Research Centre
Edmonton, AB T6G 2S2
Canada

Dear Dr. Riddell,

Thank you for submitting your manuscript entitled "Loss of polarity regulators initiates gasdermin E mediated pyroptosis in human maternal fetal interface trophoblasts" to Life Science Alliance. The manuscript was assessed by expert reviewers, whose comments are appended to this letter. We invite you to submit a revised manuscript addressing the Reviewer comments.

Thank you for this interesting contribution to Life Science Alliance. We are looking forward to receiving your revised manuscript.

Sincerely,

B. MANUSCRIPT ORGANIZATION AND FORMATTING:

Reviewer #1 (Comments to the Authors (Required)):

In this study, the authors investigated syncytiotrophoblast (ST) polarity to determine whether atypical protein kinase-c (aPKC) isoforms were implicated in ST dysfunction leading to loss of microvilli. The authors used placental explants from early pregnancy and term deliveries together with molecular and imaging techniques to demonstrate that loss of aPKC expression or function resulted in ST pyroptosis mediated by gasdermin E. To establish a triggering mechanism for this pathway, the authors used TNF and induced isoform-specific decreases in aPKC and pyroptosis. The authors therefore implicate aPKCs as important mediators of ST function and suggest that dysregulation of these molecules can lead to cell death that may participate in placental dysfunction leading to pregnancy complications.

Overall, the research question is interesting, the methodology seems appropriate, and the results are presented well. Specific comments are as follows:

1. A notable limitation of the study is the lack of demonstration that the observed mechanism takes place in placentas from complicated pregnancies. The authors touch on this slightly, but some proof-of-concept would substantially enhance the value of the manuscript.
2. Introduction: The third paragraph could use some revision for concision and fluidity, as many molecules/abbreviations are introduced rapidly, and it can be a little difficult to follow for an unfamiliar reader.
3. Results: How does the colocalization coefficient compare with more widely used quantitative or semi-quantitative methods? Does this value reflect changes in signal strength/positive cells, or only colocalization?
4. Results: Figure 2A shows an increase in green signal (p-Ezrin) within the tissue, rather than on the surface, after aPKC inhibitor treatment. This observation does not seem to be addressed in the results; is it expected/meaningful?
5. Results: To support the statement "Importantly, this strong diffuse pattern of dextran uptake was restricted to areas of the ST lacking a visible continuous apical phalloidin signal", do the authors have any images (perhaps slightly lower magnification) showing the inconsistent dextran uptake?
6. An interesting observation was the change in endocytosis and stalled endosomal trafficking with aPKC. Could the authors further discuss what changes they might expect in this regard? This is particularly important as the shedding of placental extracellular vesicles into the maternal circulation is thought to act as a means of fetal-maternal signaling.
7. Results: Did the authors test or consider other acute-phase cytokines (besides TNF) that have been implicated in pregnancy complications, such as IL-6 or IL-1 β ? The authors measure IL-1 β as an outcome of aPKC or TNF treatment, but it seems that it was not tested as the primary stimulant. Given that pyroptosis can involve inflammasome activation, it is interesting that IL-1 β was released only by first trimester conditioned medium, particularly when considering that other groups have demonstrated NLRP3/IL-1 β involvement in late pregnancy complications such as preterm labor and birth.
8. Discussion: The authors suggest that their findings have implications in pregnancy disorders initiated in early gestation (i.e., early-onset preeclampsia) as well as infection, which is generally associated with late gestation (i.e., preterm labor and birth). Do the authors consider that the mechanism displayed here is relevant for and occurring in both preeclampsia and infection-induced complications?

Reviewer #2 (Comments to the Authors (Required)):

This manuscript by Patel et al constitutes a well written and comprehensive study that fits well within the scope of the journal. The authors report on the co-localization of aPKC components and Ezrin at the apical surface of human syncytiotrophoblast from the 7th week of gestation onwards, and how this cell polarity is regulated by TNF α and Gasdermin E, a pyroptosis marker. The presented data are fundamental, novel, and exciting. This information may be of high relevance in future studies of placentas from pregnancy complications, which often are characterized by a dysfunctional syncytiotrophoblast layer. The manuscript requires some improvements on the following points:

1. For many of the confocal images of explants, it is unclear what the tissue structure is or what indeed is labelled. All these figures require brightfield or DAPI images for orientation, so to be able to co-localize signals with the anatomy of the tissue and tissue architecture.
2. In Figure 1C, bottom panel, it appears that individual cell outlines are labelled. What are these cells, and where is the

syncytial layer?

3. Many of the images are blurry, even though they were presumably taken by confocal microscopy. Prominent examples are Ezrin in Fig 1A, top row; phalloidin in Fig. 2F etc. Please carefully go through all the images and provide sharper examples as needed.
4. Many of the images are also extremely faint. Please consistently increase the brightness throughout.
5. Multiple examples of arrowheads (or similar annotations) are not described in the legends.
6. The shRNA KD examples of Western blots in Fig. 2 are not very convincing, in particular 2I. It is difficult to believe that the quantification values from these blots are so discrete. Please provide better examples of these blots.
7. Why did the authors exclusively focus on pyroptosis, while not even considering apoptosis or other cell death pathway to be involved? It would be valuable to investigate at least one additional apoptosis marker to corroborate the emphasis on the highlighted cell death pathway.
8. How is IL1b regulated over gestation, and could this explain why its release is not significant in third trimester explants?
9. If journal style permits, the Results section would be greatly improved by having subheadings.

Minor comments:

- Line 22: It is misleading to describe the ST as the "maternal surface" since the tissue is fetally derived; this should be rephrased to "maternal blood-facing surface" or similar.
- Please describe the functional difference between Ezrin and p-Ezrin
- Line 32: Please add a brief explanation of "pyroptosis" in the abstract as one form of cell death.
- Line 40: The placenta is a hybrid organ comprised of maternally derived decidua and fetal components. Its description as "fetal" organ is misleading.
- Line 101: Please rephrase, Ezrin is not an ST "marker" (which would imply some specificity for ST), it is merely labelling the apical surface of ST as one of many polarized epithelia.
- Line 113: Sentence is too long. Please consider splitting it in two.
- Lines 246-250: Sentence is unclear, please rephrase.
- Line 174: Please explain what transferrin and EEA1 are markers of.
- Line 211: "induces".
- Line 220: "...but not IN term explant..."
- Line 289: Long sentence, consider splitting into two.
- Line 302: "IL-1 β " instead of "IL- β "

We would like to thank the reviewers for their overall positive feedback on our work and their insightful and helpful comments. We have extensively revised the manuscript accordingly. The changes and additions suggested by the reviewers have substantially enhanced the readability of the manuscript and helped solidify our conclusions.

We believe that our manuscript clearly shows that disruption of the aPKC isoforms or exposure to TNF- α in syncytiotrophoblasts leads to the induction of gasdermin-E dependent pyroptosis. To the best of our knowledge this is the first work to clearly delineate that the human syncytium undergoes a form of programmed necrosis. Therefore, we believe this work answers a key fundamental problem in our field: how does the syncytiotrophoblast die? We hope that the reviewers agree that this has widespread implications for placental biology and that they are satisfied with the quality and breadth of our results and discussion. Please see the responses to the individual points raised below.

Reviewer #1:

In this study, the authors investigated syncytiotrophoblast (ST) polarity to determine whether atypical protein kinase-c (aPKC) isoforms were implicated in ST dysfunction leading to loss of microvilli. The authors used placental explants from early pregnancy and term deliveries together with molecular and imaging techniques to demonstrate that loss of aPKC expression or function resulted in ST pyroptosis mediated by gasdermin E. To establish a triggering mechanism for this pathway, the authors used TNF and induced isoform-specific decreases in aPKC and pyroptosis. The authors therefore implicate aPKCs as important mediators of ST function and suggest that dysregulation of these molecules can lead to cell death that may participate in placental dysfunction leading to pregnancy complications.

Overall, the research question is interesting, the methodology seems appropriate, and the results are presented well. Specific comments are as follows:

- 1. A notable limitation of the study is the lack of demonstration that the observed mechanism takes place in placentas from complicated pregnancies. The authors touch on this slightly, but some proof-of-concept would substantially enhance the value of the manuscript.***

We agree with the reviewer that identifying if aPKC levels are altered and pyroptosis is occurring in the ST in established pregnancy complications is an exciting direction for the future which we intend to pursue. The best way we presently have to identify when gasdermin-E mediated pyroptosis is occurring is dextran uptake assays, therefore this requires the collection of fresh, unfixed placental samples. With COVID-19 related restrictions in place for the past 3 years we have been unable to collect any of these samples to perform uptake assays. We have preliminary data from pre-COVID pandemic samples that suggest that aPKC isoform expression is altered in placental dysfunction (Figure 1 below). Based on this preliminary data and the frequency of deliveries at our site to access fresh tissue we estimate that it would take 3-5 years to obtain enough samples to accurately power these experiments. Therefore, we have not included complicated pregnancy samples in the manuscript.

We believe that our current data, in particular the work with first trimester samples, suggests that the initiation of pyroptosis in the syncytiotrophoblast may be an important driver in the development of pathology, and that pyroptosis may be occurring before the onset of maternal symptoms. To prove this hypothesis will require *in vivo* models and these experiments will also be carried out in the future. But, we believe that the sum of our observations in this manuscript still substantially pushes the field of syncytiotrophoblast programmed cell death forward, and that the involvement of pyroptosis in the pathogenesis of placental complications is an exciting and important future topic of research.

[Figure removed by editorial staff per author's request].

2. Introduction: The third paragraph could use some revision for concision and fluidity, as many molecules/abbreviations are introduced rapidly, and it can be a little difficult to follow for an unfamiliar reader.

Please see the revised lines 72-93. We hope that the reviewer finds this section more accessible now.

3. Results: How does the colocalization coefficient compare with more widely used quantitative or semi-quantitative methods? Does this value reflect changes in signal strength/positive cells, or only colocalization?

This value represents the signal overlap between two channels and colocalization (please see PMID: [21209361](https://pubmed.ncbi.nlm.nih.gov/21209361/) for an extended explanation of how this is used in image analysis). Therefore, it indicates spatial proximity of the signals irrespective of their signal intensity so long as they are above the background threshold. We apologize but it is unclear from the comment what other methods this reviewer may be referring to for spatial proximity analyses and which

experiments they may be questioning the use of this method in for us to further explain why we selected this method.

4. Results: Figure 2A shows an increase in green signal (p-Ezrin) within the tissue, rather than on the surface, after aPKC inhibitor treatment. This observation does not seem to be addressed in the results; is it expected/meaningful?

We believe this signal is p-Ezrin localized in the cytotrophoblasts. Cytotrophoblasts can also express ezrin and therefore have p-Ezrin (PMID: 25052094). We observed an inconsistent signal for both p-Ezrin and Ezrin in the cytotrophoblasts during our study. To avoid confusion for the readers we have changed the control panel to better match the images between treatments so that both cytotrophoblasts and syncytiotrophoblast apical surface can be seen in both conditions and added the reference that ezrin is also expressed in the cytotrophoblasts to the text (see line 110-111).

5. Results: To support the statement "Importantly, this strong diffuse pattern of dextran uptake was restricted to areas of the ST lacking a visible continuous apical phalloidin signal", do the authors have any images (perhaps slightly lower magnification) showing the inconsistent dextran uptake?

We have now included additional data panels in Figure 3C and Figure S8C that are tile-scanned image composites of an aPKC inhibitor and a TNF- α treated explant showing the regionalized effect on dextran uptake to areas with diminished apical phalloidin signal. These images are a composite of individual image panels and cover a 200 μ m x 400 μ m area (aPKC inhibitor treated) and a 300 μ m x 400 μ m area (TNF- α treated) showing that dextran uptake is regionalized within the ST and that dextran signals are highest in regions of the ST lacking apical phalloidin signal or with diminished phalloidin intensity.

6. An interesting observation was the change in endocytosis and stalled endosomal trafficking with aPKC. Could the authors further discuss what changes they might expect in this regard? This is particularly important as the shedding of placental extracellular vesicles into the maternal circulation is thought to act as a means of fetal-maternal signaling.

We agree that this was a very interesting observation, and after further literature searches now hypothesize that the enlarged endosomes are likely to be a result of a membrane damage repair response (please see PMID: PMC6420883 for a review on the topic) and not due to stalled endosomal maturation. We have included a new section in the discussion specifically focused on the enlarged endosomes and how pore-forming proteins can elicit both endo and exocytosis (lines 366-387) and we are currently investigating membrane damage-induced responses in the ST in a new study.

7. Results: Did the authors test or consider other acute-phase cytokines (besides TNF) that have been implicated in pregnancy complications, such as IL-6 or IL-1 β ? The authors measure IL-1 β as an outcome of aPKC or TNF treatment, but it seems that it was not tested as the primary stimulant. Given that pyroptosis can involve inflammasome activation, it is interesting that IL-1 β was released only by first trimester conditioned medium, particularly when considering that other groups have demonstrated NLRP3/IL-1 β involvement in late pregnancy complications such as preterm labor and birth.

In addition to testing TNF- α we have preliminary work treating isolated trophoblasts with the acute phase cytokine IFN- γ . As is shown in the provided western blot (Figure 2 below), we see no evidence of altered aPKC expression after stimulation with IFN- γ , suggesting there is specificity in the ability of cytokines to regulate aPKC abundance and presumably the initiation of pyroptosis. At this time, we have not investigated whether IL-1 β or IL-6 can also initiate pyroptosis. We agree with this reviewer that this is a very interesting future direction, and feel that testing of multiple potential initiating factors, such as infection, individual cytokines, co-administered cytokine cocktails, and other DAMPS will be important for understanding when this death pathway is initiated, and we intend to carry out these experiments in the future.

[Figure removed by editorial staff per author's request].

As for the IL-1 β release significantly rising only in the first trimester explants with aPKC inhibitor and TNF- α treatment, this was also brought up by Reviewer 2, and we feel is a very interesting observation. Please also see our response below (point 8) about the potential gestational age effects on IL-1 β expression and processing itself. Gasdermin pores are just one mechanism through which IL-1 β can be released from a cell (PMID: 37090724, 30089254) therefore one possibility is that there are different pathways regulating IL-1 β release between first trimester and term trophoblasts, with GSDME-mediated release predominating in first trimester samples in response to acute stimuli and an alternative pathway being responsible for acute IL-1 β release at term. Importantly, there were detectable levels of IL-1 β in both first trimester and term explant supernatants, just the increased release of IL-1 β after treatment was not observed in term tissue despite permeabilization of the ST to dextran.

Regarding the involvement of the NLRP3 inflammasome in gasdermin-E dependent pyroptosis and IL-1 β release. We have preliminary data showing that gasdermin-E dependent dextran uptake in first trimester ST is independent of NLRP3 activation (Figure 3 below), therefore we hypothesize that the NLRP3 inflammasome is important for the maturation of IL-1 β but whether it is important for regulating the pathways that mediate its release remains to be tested. Interestingly, NLRP3 inhibitor treatment alone led to increased dextran uptake in the first trimester ST (Figure 3 below), thus there is unexpected complexity in downstream effects of NLRP3 activation in ST that remains to be identified. We have not tested the NLRP3 inhibitor in term ST yet to know whether similar effects would be seen in older tissue. But taken together, we believe that the provided preliminary data and the data in our

manuscript highlight that much more work needs to be carried out to understand the up- and downstream effects of NLRP3 activation at different points in gestation and how these effect IL-1 β dynamics.

[Figure removed by editorial staff per author's request].

8. Discussion: The authors suggest that their findings have implications in pregnancy disorders initiated in early gestation (i.e., early onset preeclampsia) as well as infection, which is generally associated with late gestation (i.e., preterm labor and birth). Do the authors consider that the mechanism displayed here is relevant for and occurring in both preeclampsia and infection-induced complications?

Yes. Our data with TNF- α would suggest that conditions that lead to high local TNF- α concentrations in the vicinity of the ST could lead to the initiation of pyroptosis. We believe that pyroptosis is therefore likely to be activated in a wide variety of situations, similar to how apoptosis is a commonly activated mechanism to stimulate tissue turnover and as part of a tissue-mediated inflammatory response in other epithelia. We have added a section to the discussion about how pore-forming proteins, like gasdermin-D, induce a membrane repair mechanism in cells that results in the increased export of extracellular vesicles, in addition to our data showing ST membrane blebbing leading to the release of membranous vesicles after TNF- α treatment. The increased release of ST derived extracellular vesicles is hypothesized to be a primary driver in the development of maternal endothelial cell dysfunction in preeclampsia. Therefore, chronic initiation of ST pyroptosis may directly contribute to the development maternal endothelial cell dysfunction.

Interestingly, murine placental zika virus infection has been shown to lead to placental gasdermin-E mediated pyroptosis in a TNF- α dependent manner (PMID 35972780), suggesting that placental infection can indirectly stimulate pyroptosis. Thus, highlighting that pyroptosis may also play a role in placental response to infection.

Reviewer 2:

1. For many of the confocal images of explants, it is unclear what the tissue structure is or what indeed is labelled. All these figures require brightfield or DAPI images for orientation, so to be able to co-localize signals with the anatomy of the tissue and tissue architecture.

We appreciate that the complex structure of the tissue makes it difficult to interpret where the ST is and the position of the underlying cells. Unfortunately, it is not possible to include a nuclear stain in all our images because the confocal microscope we have can only support three colour imaging. Additionally, for some experiments there are no nuclear dyes compatible with the detectors and lasers we have on our confocal microscope that would not bleed through and affect the quantitation of our channels of interest (since nuclear dyes are inherently very highly fluorescent and we do not have a far red laser/detector). Similarly, we cannot produce bright field overlays for the images due to the limitations of our system. To address the issue raised here we have added Figures 1A and 1B which introduce the reader to the unique actin-cytoskeletal structure of the ST using representative images from late-first trimester tissue and term tissue to help educate a reader how to interpret the presented image views. We hope the reviewer feels this is instructive enough to allow readers to be able to interpret our images.

2. In Figure 1C, bottom panel, it appears that individual cell outlines are labelled. What are these cells, and where is the syncytial layer?

In the former Figure 1C the cells that were outlined are likely to be cytotrophoblasts, which we have previously shown also express aPKC- ζ isoforms (PMID:35124330). The additional lines were from infolds of the ST surface. Our images are taken from whole mount tissue therefore the 3D shape of the ST is very well maintained when we examine 3D reconstituted z-stack images. Since the presented images are a single focal plane of z-stack confocal images and the ST surface is not flat these fold points can look like cells but when the YZ image plane is examined they are contiguous with the apical surface of the ST. To make this image easier for the reader to interpret we have replaced it with a new image (Figure 1E, bottom) to help the reader to identify where the ST apical and basal surfaces are. We hope this combined with the new orientation figure make it easier for the reader to interpret the images.

3. Many of the images are blurry, even though they were presumably taken by confocal microscopy. Prominent examples are Ezrin in Fig 1A, top row; phalloidin in Fig. 2F etc. Please carefully go through all the images and provide sharper examples as needed.

We have now replaced image 1A as requested and carefully check the images. We apologize but some of the blurriness in images (for example Figure 2F) were from mistakenly inserted lower quality JPEG images when preparing the figures. We hope the reviewer is satisfied with the improved images.

4. Many of the images are also extremely faint. Please consistently increase the brightness throughout.

We have gone through the manuscript and increased the brightness on many of the image panels as requested.

5. Multiple examples of arrowheads (or similar annotations) are not described in the legends.

We have carefully gone through the manuscript and identified these omissions. We are sorry for these mistakes in the original submission. We hope we have corrected them all.

6. The shRNA KD examples of Western blots in Fig. 2 are not very convincing, in particular 2I. It is difficult to believe that the quantification values from these blots are so discrete. Please provide better examples of these blots.

We have included a new representative blot for the siRNA knockdown in Figure 2I. We hope that the reviewer is satisfied with this new blot. Also, please note that the graphs represent values normalized to the controls from the individual experiments (donors), as we have seen a wide degree in variation in the amount of aPKC isoforms/patient sample, hence the very tight values in the summary data. In addition, we would like to bring to the reviewer's attention that this method of siRNA delivery results in the accumulation of siRNA selectively in the ST (please see PMID: 19012963) with limited delivery of siRNA to the underlying CT. Both the ST and CT express all three aPKC isoforms (PMID:35124330), therefore we believe that the relatively modest knockdown efficiencies are due to the remaining protein that are primarily expressed in the CT. Since our dextran and F-actin abundance results with aPKC inhibitor treatment and siRNA knockdown with two different sets of siRNAs for aPKCs are complimentary, we believe that we are achieving a significant degree of aPKC isoform knockdown in the ST.

7. Why did the authors exclusively focus on pyroptosis, while not even considering apoptosis or other cell death pathway to be involved? It would be valuable to investigate at least one additional apoptosis marker to corroborate the emphasis on the highlighted cell death pathway.

We did not extensively consider apoptosis due to the convincing evidence presented by Longtine *et. al* (PMCID: PMC3631347) showing that the ST does not undergo apoptosis. Our data has corroborated their observations, since we also found that active caspase-3 signal is restricted to cells in the placental core and is not observed in the ST. We have also never observed ST nuclear fragmentation in aPKC nor TNF- α treated tissue. Additionally, our scanning electron microscopy images appeared so strikingly similar to those of Herr *et. al* in pyroptotic microglia (PMCID: PMC7230060) we pursued this direction. Unfortunately, almost no available cell death assays that we could identify (for example TUNEL assays) can clearly distinguish between an apoptotic and pyroptotic/necrotic cell (PMCID: PMC8005494), particularly because the ST appears to be executing a non-canonical pyroptotic cascade. Since one clear difference between apoptotic and pyroptotic cells is the occurrence of nuclear disassembly (PMCID: PMC8005494), we have included in lines 244-250 this distinction between the death pathways and our observations that nuclear fragmentation can be observed in the stromal cells, but not within the ST and example images highlighting nuclear fragmentation in stromal cells are now included in Figure S9A.

8. How is IL-1 β regulated over gestation and could this explain why its release is not significant in third trimester explants?

Other groups have observed that stimulation of term placentas with LPS leads to the suppression of IL-1 β release into the maternal compartment in placental perfusion model

(PMID: 18471873), whereas this is not the case in preterm placentas where LPS stimulates IL-1 β release in the maternal compartment. Therefore, we believe that there may be different pathways regulating the release mechanisms of IL-1 β in a gestational age-dependent manner (as discussed above in response to reviewer #1's point 7). We have not been able to find any clear data in the literature comparing the gestational age-dependent expression of IL-1 β , nor caspase-1, which is necessary for its maturation and secretion in trophoblasts. There is a great deal of literature looking at the differences in circulating levels of or mRNA levels of IL-1 β between control and pathologic pregnancies, but a paucity of data about gestational age effects. Therefore, we agree with this reviewer, that it could be that there are differences in the regulation of IL-1 β expression itself that contribute to our results, or that there are gestational age-dependent changes in the regulation of its release. We will investigate all these ideas thoroughly in the future.

9. If journal style permits, the Results section would be greatly improved by having subheadings.

Thank you for this suggestion. Subheadings have now been included.

Minor comments:

Line 22: It is misleading to describe the ST as the "maternal surface" since the tissue is fetally derived; this should be rephrased to "maternal blood-facing surface" or similar.
Completed.

Please describe the functional difference between Ezrin and p-Ezrin

Please see lines 63-66 and 143-146.

Line 32: Please add a brief explanation of "pyroptosis" in the abstract as one form of cell death.

Please see line 35.

Line 40: The placenta is a hybrid organ comprised of maternally derived decidua and fetal components. Its description as "fetal" organ is misleading.

We respectfully disagree with the reviewer. The placenta is a separate tissue from the decidua, though intimately associated with it. The endometrium begins to decidualize in the latter half of the menstrual cycle before implantation in humans (PC MID: PMC7312091), therefore decidua can exist without a placenta and in our opinion constitutes a distinct tissue. The cells that make up the placenta are fetally encoded and derived, hence why we feel it is important to include that it is a fetal organ. We believe that it causes confusion for readers less familiar with the structure of the human maternal/fetal interface to not clearly explain that placental cells are produced by the embryo alone. In the context of our work here we believe this is important to point out for less expert readers to highlight that pyroptosis leads to the release of fetally encoded components into the maternal system.

Line 101: Please rephrase, Ezrin is not an ST "marker" (which would imply some specificity for ST), it is merely labelling the apical surface of ST as one of many polarized epithelia.

We have rephrased this sentence to avoid confusion, as we intended to say exactly what the reviewer did, that ezrin is a marker of the apical membrane (current line 109-110).

Line 113: Sentence is too long. Please consider splitting it in two.
Complete.

Lines 246-250: Sentence is unclear, please rephrase.
Complete.

Line 174: Please explain what transferrin and EEA1 are markers of.
Complete. Please see current lines 197-201.

Line 211: "induces".
Fixed.

Line 220: "...but not IN term explant..."
Fixed.

Line 289: Long sentence, consider splitting into two.
Fixed.

Line 302: "IL-1 β " instead of "IL- β "
Fixed.

July 6, 2023

RE: Life Science Alliance Manuscript #LSA-2023-01946-TR

Dr. Meghan Riddell
University of Alberta
Obstetrics and Gynecology
227 Heritage Medical Research Centre
Edmonton, AB T6G 2S2
Canada

Dear Dr. Riddell,

Thank you for submitting your revised manuscript entitled "Loss of polarity regulators initiates gasdermin-E mediated pyroptosis in syncytiotrophoblasts". We would be happy to publish your paper in Life Science Alliance pending final revisions necessary to meet our formatting guidelines.

- please upload your Table in editable .doc or Excel format
- please upload all figure files as individual ones, including the supplementary figure files; all figure legends should only appear in the main manuscript file
- please cite Table 1 in the manuscript text
- please add a callout for Figure S3A,B; S5A-H; S8C; S10A-C; S12A-B; S13A-B to your main manuscript text;

A. FINAL FILES:

B. MANUSCRIPT ORGANIZATION AND FORMATTING:

Sincerely,

Reviewer #1 (Comments to the Authors (Required)):

Thank you for thoroughly addressing all of my comments.

Reviewer #2 (Comments to the Authors (Required)):

The authors have comprehensively addressed all issues that were raised in the previous review. The manuscript and in particular the clarity of the images is much improved.

July 12, 2023

RE: Life Science Alliance Manuscript #LSA-2023-01946-TRR

Dr. Meghan Riddell
University of Alberta
Obstetrics and Gynecology
227 Heritage Medical Research Centre
Edmonton, AB T6G 2S2
Canada

Dear Dr. Riddell,

Thank you for submitting your Research Article entitled "Loss of polarity regulators initiates gasdermin-E mediated pyroptosis in syncytiotrophoblasts". It is a pleasure to let you know that your manuscript is now accepted for publication in Life Science Alliance. Congratulations on this interesting work.

DISTRIBUTION OF MATERIALS:

Again, congratulations on a very nice paper. I hope you found the review process to be constructive and are pleased with how the manuscript was handled editorially. We look forward to future exciting submissions from your lab.

Sincerely,
